# Prior and Posterior Networks: A Survey on Evidential Deep Learning Methods For Uncertainty Estimation

## Abstract

Popular approaches for quantifying predictive uncertainty in deep neural networks often involve multiple sets of weights or models, for instance, via ensembling or Monte Carlo dropout. These techniques usually incur overhead by having to train multiple model instances or do not produce very diverse predictions. This survey aims to familiarize the reader with an alternative class of models based on the concept of *Evidential Deep Learning*: For unfamiliar data, they admit "what they don't know" and fall back onto a prior belief. Furthermore, they allow uncertainty estimation in a single model and forward pass by parameterizing *distributions over distributions*. This survey recapitulates existing works, focusing on the implementation in a classification setting, before surveying the application of the same paradigm to regression. We also reflect on the strengths and weaknesses compared to each other as well as to more established methods and provide the most central theoretical results using a unified notation in order to aid future research.

## 1 Introduction

Many existing methods for uncertainty estimation leverage the concept of Bayesian model averaging, which approaches such as Monte Carlo (MC) dropout (Gal & Ghahramani, 2016), Bayes-by-backprop (Blundell et al., 2015) or ensembling (Lakshminarayanan et al., 2017) can be grouped under (Wilson & Izmailov, 2020). This involves the approximation of an otherwise infeasible to compute integral using MC samples — for instance from an auxiliary distribution or in the form of ensemble members. This causes the following problems: Firstly, the quality of the MC approximation depends on the veracity and diversity of samples from the weight posterior. Secondly, the approach often involves increasing the number of parameters in a model or training more model instances altogether. Recently, a new class of models has been proposed to side-step this conundrum by using a different factorization of the posterior predictive distribution. This allows computing uncertainty in a single forward pass and a single set of weights. Furthermore, these models are grounded in a concept coined *Evidential Deep Learning*: For out-of-distribution (OOD) inputs, they fall back onto a prior, often expressed as *knowing what they don't know.*

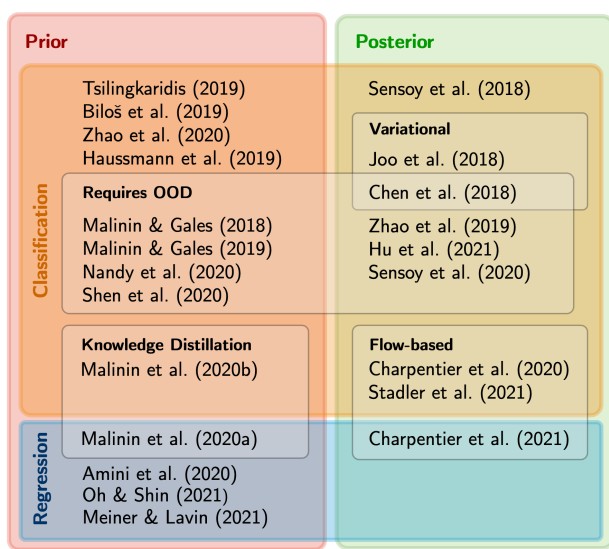

Figure 1: Taxonomy of surveyed approaches. For more detail about prior and posterior networks for classification, check Tables 1 and 2, respectively. See Table 3 for all regression approaches.

In this paper, we summarize the existing literature and group Evidential Deep Learning approaches, critically reflecting on their advantages and shortcomings. This survey aims to both serve as an accessible introduction to this model family to the unfamiliar reader as well as an informative overview, in order to promote a wider application outside the uncertainty estimation literature. We also provide a collection of the most important theoretical results for the Dirichlet distribution for Machine Learning, which plays a central role in many of the discussed approaches. We give an overview over all discussed work in Figure 1, where we distuinguish surveyed works for classification between models parameterizing a Dirichlet prior (Section 3.3.1) or posterior (Section 3.3.2). We further discuss similar methods for regression problems (Section 4). As we will see, obtaining well-behaving uncertainty estimates can be challenging in the Evidential Deep Learning framework; proposed solutions that are also reflected in Figure 1 are the usage of OOD examples during training (Malinin & Gales, 2018; 2019; Nandy et al., 2020; Shen et al., 2020; Chen et al., 2018; Zhao et al., 2019; Hu et al., 2021; Sensoy et al., 2020), knowledge distillation (Malinin et al., 2020b;a) or the incorporation of density estimation (Charpentier et al., 2020; 2021; Stadler et al., 2021), which we discuss in more detail in Section 6.

## 2 Background

We first introduce the central concepts to this survey, including Bayesian model averaging in Section 2.2 and Evidential Deep Learning in Section 2.3, along with a short introduction to the Dirichlet distribution we use to illustrate the relevant concepts behind Bayesian inference.

### 2.1 The Dirichlet distribution

Underlying the following sections is the concept of Bayesian inference: Given some prior belief $p(\boldsymbol{\theta})$ about parameters of interest $\boldsymbol{\theta}$, we use available observations $\mathbb{D}$ and their likelihood $p(\mathbb{D}|\boldsymbol{\theta})$ to obtain an updated belief in form of the posterior $p(\boldsymbol{\theta}|\mathbb{D}) \propto p(\mathbb{D}|\boldsymbol{\theta})p(\boldsymbol{\theta})$. Within Bayesian inference, the Beta distribution is a commonly used prior for a Bernoulli likelihood, which can e.g. be used in a binary classification problem. Whereas the Bernoulli likelihood has a single parameter $\mu$, indicating the probability of success (or of the positive class), the Beta distribution posses two shape parameters $\alpha_1$ and $\alpha_2$, and is defined as follows:

$$\text{Beta}(\mu; \alpha_1, \alpha_2) = \frac{1}{B(\alpha_1, \alpha_2)} \mu^{\alpha_1 - 1}(1 - \mu)^{\alpha_2 - 1}; \quad B(\alpha_1, \alpha_2) = \frac{\Gamma(\alpha_1)\Gamma(\alpha_2)}{\Gamma(\alpha_1 + \alpha_2)}; \tag{1}$$

where $\Gamma(\cdot)$ stands for the gamma function, a generalization of the factorial to the real numbers, and $B(\cdot)$ is called the Beta function (not to be confused with the distribution). As such, the Beta distribution expresses a *probability over probabilities*: That is, the probability of different parameter values for $\mu$. The Dirichlet distribution arises as a multivariate generalization of the Beta distribution (and is thus also called the *multi-variate Beta distribution*) for a multi-class classification problem and is defined as follows:

$$\text{Dir}(\boldsymbol{\mu}; \boldsymbol{\alpha}) = \frac{1}{B(\boldsymbol{\alpha})} \prod_{k=1}^{K} \mu_k^{\alpha_k - 1}; \quad B(\boldsymbol{\alpha}) = \frac{\prod_{k=1}^{K} \Gamma(\alpha_k)}{\Gamma(\alpha_0)}; \quad \alpha_0 = \sum_{k=1}^{K} \alpha_k; \quad \alpha_k \in \mathbb{R}^+; \tag{2}$$

where $K$ denotes the number of categories or classes, and the Beta function $B(\cdot)$ is now defined for $K$ shape parameters compared to Equation (1). For notational convenience, we also define $\mathbb{K} = \{1, \dots, K\}$ as the set of all classes. The distribution is characterized by its *concentration parameters* $\boldsymbol{\alpha}$, the sum of which, often denoted as $\alpha_0$, is called the *precision*.[1] The distribution becomes relevant for applications using neural networks, considering that most neural networks for classification use a softmax function after their last layer to produce a categorical distribution of classes $\text{Cat}(\boldsymbol{\mu}) = \prod_{k=1}^{K} \mu_k^{\mathbf{1}_{y=k}}$, in which the class probabilities are expressed using a vector $\boldsymbol{\mu} \in [0,1]^K$ s.t. $\mu_k \equiv P(y = k|x)$ and $\sum_k \mu_k = 1$. The Dirichlet is a *conjugate prior* for such a categorical likelihood, meaning that in combination they produce a Dirichlet posterior with parameters $\boldsymbol{\beta}$, given a data set $\mathbb{D} = \{(x_i, y_i)\}_{i=1}^{N}$ of $N$ observations with corresponding labels:

---

[1]The precision is analogous to the precision of a Gaussian, where a larger $\alpha_0$ signifies a sharper distribution.

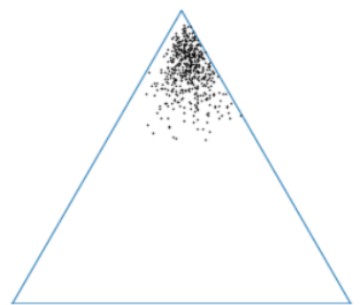

(a) Categorical distributions predicted by a neural ensemble on the probability simplex.

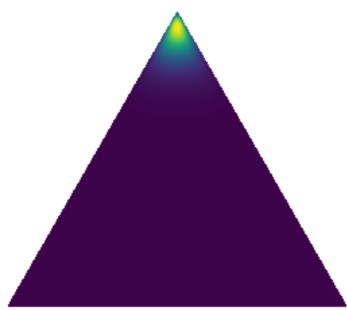

(b) Probability simplex for a confident prediction, for with the density concentrated in a single corner.

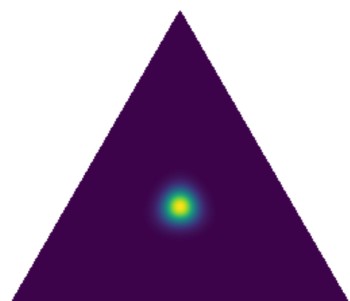

(c) Dirichlet distribution for a case of data uncertainty, with the density concentrated in the center.

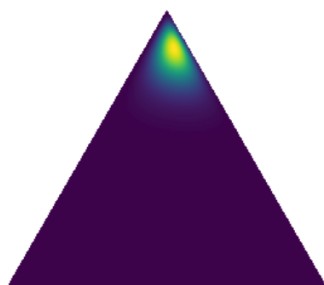

(d) Dirichlet distribution for a case of model uncertainty, with the density spread out more.

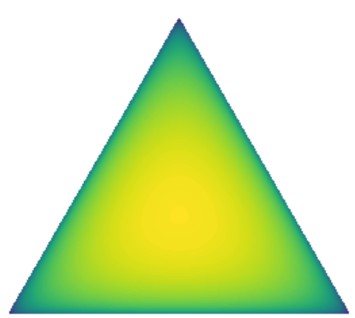

(e) Dirichlet for a case of distributional uncertainty, with the density spread across the whole simplex.

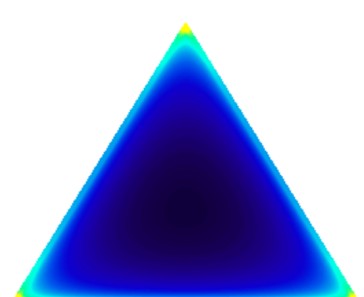

(f) Alternative approach to distributional uncertainty called representation gap, with density concentrated along the edges.

Figure 2: Examples of the probability simplex for a $K = 3$ classification problem, where every corner corresponds to a class and every point to a categorical distribution. Brighter colors correspond to higher density. (a) Predicted categorical distributions by an ensemble of discriminators. (b) – (e) (Desired) Behavior of Dirichlet in different scenarios by Malinin & Gales (2018): (b) For a confident prediction, the density is concentrated in the corner of the simplex corresponding to the assumed class. (c) In the case of aleatoric uncertainty, the density is concentrated in the center, and thus uniform categorical distributions are most likely. (d) In the case of model uncertainty, the density may still be concentrated in a corner, but more spread out, expressing the uncertainty about the right prediction. (e) In the case of an OOD input, a uniform Dirichlet expresses that any categorical distribution is equally likely, since there is no evidence for any known class. (f) Representation gap by Nandy et al. (2020), proposed as an alternative behavior for OOD data. Here, the density is instead concentrated solely on the edges of the simplex.

$$p(\boldsymbol{\mu}|\mathbb{D}, \boldsymbol{\alpha}) \propto p(\{y_i\}_{i=1}^N | \boldsymbol{\mu}, \{x_i\}_{i=1}^N) p(\boldsymbol{\mu}|\boldsymbol{\alpha}) = \prod_{i=1}^N \prod_{k=1}^K \mu_k^{\mathbf{1}_{y_i=k}} \frac{1}{B(\boldsymbol{\alpha})} \prod_{k=1}^K \mu_k^{\alpha_k - 1}$$

$$= \prod_{k=1}^K \mu_k^{\left(\sum_{i=1}^N \mathbf{1}_{y_i=k}\right)} \frac{1}{B(\boldsymbol{\alpha})} \prod_{k=1}^K \mu_k^{\alpha_k - 1} = \frac{1}{B(\boldsymbol{\alpha})} \prod_{k=1}^K \mu_k^{N_k + \alpha_k - 1} \propto \mathrm{Dir}(\boldsymbol{\mu}; \boldsymbol{\beta}),$$

$$(3)$$

where $\boldsymbol{\beta}$ is a vector with $\beta_k = \alpha_k + N_k$, with $N_k$ denoting the number of observations for class $k$ and $\mathbf{1}$ being the indicator function. Intuitively, this implies that the prior belief encoded by the initial Dirichlet is updated using the actual data, sharpening the distribution for classes for which many instances have been observed. Similar to the Beta distribution in Equation (1), the Dirichlet is a *distribution over categorical distributions*

on the $K-1$ probability simplex; multiple instances of which are shown in Figure 2. Each point on the simplex corresponds to a categorical distribution, with the proximity to a corner indicating a high probability for the corresponding class. Figure 2a displays the predictions of an ensemble of classifiers as a point cloud on the simplex. Using a Dirichlet, this finite set of distributions can be extended to a continuous density over the whole simplex. As we will see in the following sections, parameterizing a Dirichlet distribution with a neural network enables us to distinguish different scenarios using the shape of its density, as shown in Figures 2b to 2f, which we will discuss in more detail along the way.

## 2.2 Predictive Uncertainty in Neural Networks

In probabilistic modelling, uncertainty is commonly divided into aleatoric and epistemic uncertainty (Der Kiureghian & Ditlevsen, 2009; Hüllermeier & Waegeman, 2021). Aleatoric refers to the uncertainty that is induced by the data-generating process, for instance noise or inherent overlap between observed instances of classes. Epistemic is the type of uncertainty about the optimal model parameters (or even hypothesis class), reducible with an increasing amount of data, as less and less possible models become a plausible fit. These two notions resurface when formulating the posterior predictive distribution for a new data point $\mathbf{x}$:[2]

$$P(y \,|\, \mathbf{x}, \mathbb{D}) = \int \underbrace{P(y \,|\, \mathbf{x}, \boldsymbol{\theta})}_{\text{Aleatoric}} \underbrace{p(\boldsymbol{\theta} | \mathbb{D})}_{\text{Epistemic}} \, d\boldsymbol{\theta}. \tag{4}$$

Here, the first term captures the aleatoric uncertainty about the correct class based on the predicted categorical distribution, while the second one expresses uncertainty about the correct model parameters — the more data we observe, the more concentrated $p(\boldsymbol{\theta}|\mathbb{D})$ should become for reasonable parameter values for $\boldsymbol{\theta}$. Since we integrate over the entire parameter space of $\boldsymbol{\theta}$, weighting each prediction by the posterior probability of its parameters to obtain the final result, this process is referred to as *Bayesian model averaging* (BMA). For a large number of real-valued parameters $\boldsymbol{\theta}$ like in neural networks, this integral becomes intractable, and is usually approximated using Monte Carlo samples — with the aforementioned problems of computational overhead and approximation errors, motivating the approaches discussed in this survey.

## 2.3 Evidential Deep Learning

Based on the mentioned concerns, Malinin & Gales (2018) therefore propose to factorize Equation (4) further:

$$p(y \,|\, \mathbf{x}, \mathbb{D}) = \iint \underbrace{P(y | \boldsymbol{\mu})}_{\text{Aleatoric}} \underbrace{p(\boldsymbol{\mu} \,|\, \mathbf{x}, \boldsymbol{\theta})}_{\text{Distributional}} \underbrace{p(\boldsymbol{\theta} | \mathbb{D})}_{\text{Epistemic}} \, d\boldsymbol{\mu} d\boldsymbol{\theta} \approx \int P(y | \boldsymbol{\mu}) \underbrace{p(\boldsymbol{\mu} \,|\, \mathbf{x}, \hat{\boldsymbol{\theta}})}_{p(\boldsymbol{\theta}|\mathbb{D}) \approx \delta(\boldsymbol{\theta} - \hat{\boldsymbol{\theta}})} \, d\boldsymbol{\mu}. \tag{5}$$

In the last step, we replace $p(\boldsymbol{\theta}|\mathbb{D})$ by a point estimate $\hat{\boldsymbol{\theta}}$ using the Dirac delta function, i.e. a single trained neural network, to get rid of the intractable integral. Although another integral remains, retrieving the uncertainty from this predictive distribution actually has a closed-form analytical solution for the Dirichlet (see Section 3.2). The advantage of this approach is further that it allows us to distinguish uncertainty about a data point because it lies in a region of considerable class overlap (Figure 2c) from points coming from an entirely different data distribution (Figure 2e). It should be noted that restricting oneself to a point estimate of the parameters prevent the estimation of epistemic uncertainty like in earlier works through the weight posterior $p(\boldsymbol{\theta}|\mathbb{D})$, as discussed in the next section. However, there are works like Haussmann et al. (2019); Zhao et al. (2020) that combine both approaches.

**Definition** The term *Evidential Deep Learning* (EDL) originates from the work of Sensoy et al. (2018) and is based on the *Theory of Evidence* (Dempster, 1968; Audun, 2018): Within the theory, belief mass is assigned to set of possible states, e.g. class labels, and can also express a lack of evidence, i.e. a "I don't know".

---

[2]Note that the predictive distribution in Equation (4) generalizes the common case for a single network prediction where $P(y \,|\, \mathbf{x}, \boldsymbol{\theta}) \approx P(y \,|\, \mathbf{x}, \hat{\boldsymbol{\theta}})$. Mathematically, this is expressed by replacing the posterior $p(\boldsymbol{\theta}|\mathbb{D})$ by a delta distribution as in Equation (5), where all probability density rests on a single parameter configuration.

We can for instance generalize the predicted output of a neural classifier using the Dirichlet distribution, allowing to express a lack of evidence through a uniform Dirichlet (Figure 2e). This is different from a uniform categorical distribution, which does not distinguish an equal probability for all classes from the lack of evidence. For the purpose of this survey, we define Evidential Deep Learning as a family of approaches for which a neural network can fall back onto a uniform prior for unknown inputs. These networks do not parameterize likelihoods, but prior or posterior distributions instead. We will discuss their implementation for classification problems further in the next section, before discussing methods for regression in Section 4.

## 3  Dirichlet Networks

We will show in Section 3.1 how neural networks can parameterize Dirichlet distributions, while Section 3.2 reveals how such parameterization can be exploited for efficient uncertainty estimation. The remaining sections enumerate different examples from the literature parameterizing either a prior (Section 3.3.1) or posterior Dirichlet distribution (Section 3.3.2) according to Equations (2) and (3).

### 3.1  Parameterization

For a classification problem with $K$ classes, a neural classifier is usually realized as a function $f_{\boldsymbol{\theta}} : \mathbb{R}^D \to \mathbb{R}^K$, mapping an input $\mathbf{x} \in \mathbb{R}^D$ to *logits* for each class. Followed by a softmax function, this then defines a categorical distribution over classes with a vector $\boldsymbol{\mu}$ with $\mu_k \equiv p(y = k | \mathbf{x}, \boldsymbol{\theta})$. The same architecture can be used without any major modification to instead parameterize a *Dirichlet* distribution, as in Equation (2).[3] In order to classify a data point $\mathbf{x}$, a categorical distribution is created from the predicted concentration parameters of the Dirichlet as follows (this definition arises from the expected value, see Appendix B.1):

$$\boldsymbol{\alpha} = f_{\boldsymbol{\theta}}(\mathbf{x}); \quad \mu_k = \frac{\alpha_k}{\alpha_0}; \quad \hat{y} = \arg\max_{k \in \mathbb{K}} \ \mu_1, \ldots, \mu_K. \tag{6}$$

This process is very similar when parameterizing a Dirichlet posterior distribution, except that in the case of the posterior, a term corresponding to the observations per class $N_k$ in Equation (3) is added to every concentration parameter as well.

### 3.2  Uncertainty Estimation with Dirichlet Networks

Let us now turn our attention to how to estimate the different notions of uncertainty laid out in Section 2.2 within the Dirichlet framework. Although stated for the prior parameters $\boldsymbol{\alpha}$, the following methods can also be applied to the posterior parameters $\boldsymbol{\beta}$ without loss of generality.

**Data (aleatoric) uncertainty.**   For the data uncertainty, we can evaluate the expected entropy of the data distribution $p(y|\boldsymbol{\mu})$ (similar to previous works like e.g. Gal & Ghahramani, 2016). As the entropy captures the "peakiness" of the output distribution, a lower entropy indicates that the model is concentrating all probability mass on a single class, while high entropy stands for a more uniform distribution — the model is undecided about the right prediction. For Dirichlet networks, this quantity has a closed-form solution (for the full derivation, refer to Appendix C.1):

$$\mathbb{E}_{p(\boldsymbol{\mu} | \mathbf{x}, \hat{\boldsymbol{\theta}})} \left[ H \Big[ P(y|\boldsymbol{\mu}) \Big] \right] = -\sum_{k=1}^{K} \frac{\alpha_k}{\alpha_0} \left( \psi(\alpha_k + 1) - \psi(\alpha_0 + 1) \right) \tag{7}$$

where $\psi$ denotes the digamma function, defined as $\psi(x) = \frac{d}{dx} \log \Gamma(x)$, and $H$ the Shannon entropy.

---

[3]The only thing to note here is that the every $\alpha_k$ has to be strictly positive, which can for instance be enforced by using an additional ReLU function (and adding a small value, e.g. like in Sensoy et al., 2020) on the output or predicting $\log \alpha_k$ instead (Sensoy et al., 2018; Malinin & Gales, 2018).

**Model (epistemic) uncertainty.** As we saw in Section 2.2, computing the model uncertainty via the weight posterior $p(\boldsymbol{\theta}|\mathbb{D})$ like in Blundell et al. (2015); Gal & Ghahramani (2016); Smith & Gal (2018) is usually not done in the Dirichlet framework.[4] Nevertheless, a key property of Dirichlet networks is that epistemic uncertainty is expressed in the spread of the Dirichlet distribution (for instance in Figure 2 (d) and (e)). Therefore, the epistemic uncertainty can be quantified considering the concentration parameters $\boldsymbol{\alpha}$ that shape this very same distribution: Charpentier et al. (2020) simply consider the maximum $\alpha_k$ as a score akin to the maximum probability score by Hendrycks & Gimpel, while Sensoy et al. (2018) compute it by $K/\sum_{k=1}^{K}(\alpha_k+1)$ or simply $\alpha_0$ (Charpentier et al., 2020). In both cases, the underlying intuition is that larger $\alpha_k$ produce a sharper density, and thus indicate increased confidence in a prediction.

**Distributional uncertainty.** Another appealing property of this model family is being able to distinguish uncertainty due to model underspecification (Figure 2d) from uncertainty due to alien inputs (Figure 2e). In the Dirichlet framework, the distributional uncertainty can be quantified by computing the difference between the total amount of uncertainty and the data uncertainty, which can be expressed in terms of the mutual information between the label $y$ and its categorical distribution $\boldsymbol{\mu}$:

$$I\Big[y, \boldsymbol{\mu}\Big|\mathbf{x}, \mathbb{D}\Big] = \underbrace{H\Big[\mathbb{E}_{p(\boldsymbol{\mu}|\mathbf{x},\mathbb{D})}\Big[P(y|\boldsymbol{\mu})\Big]\Big]}_{\text{Total Uncertainty}} - \underbrace{\mathbb{E}_{p(\boldsymbol{\mu}|\mathbf{x},\mathbb{D})}\Big[H\Big[P(y|\boldsymbol{\mu})\Big]\Big]}_{\text{Data Uncertainty}} \tag{8}$$

Given that $\mathbb{E}[\mu_k] = \frac{\alpha_k}{\alpha_0}$ (Appendix B.1) and assuming the point estimate $p(\boldsymbol{\mu}|\mathbf{x},\mathbb{D}) \approx p(\boldsymbol{\mu}|\mathbf{x},\hat{\boldsymbol{\theta}})$ to be sufficient (Malinin & Gales, 2018), we obtain an expression very similar to Equation (7):

$$I\Big[y, \boldsymbol{\mu}\Big|\mathbf{x}, \mathbb{D}\Big] = -\sum_{k=1}^{K}\frac{\alpha_k}{\alpha_0}\left(\log\frac{\alpha_k}{\alpha_0} - \psi(\alpha_k+1) + \psi(\alpha_0+1)\right) \tag{9}$$

This quantity expresses how much information we would receive about $\boldsymbol{\mu}$ if we were given the label $y$, conditioned on the new input $\mathbf{x}$ and the training data $\mathbb{D}$. In regions in which the model is well-defined, receiving $y$ should not provide much new information about $\boldsymbol{\mu}$ — and thus the mutual information would be low. Yet, such knowledge should be very informative in regions in which few data have been observed, and there the mutual information would indicate higher model uncertainty.

**Note on epistemic uncertainty estimation** The introduction of distributional uncertainty, a notion that is non-existent in the Bayesian Model Averaging framework, warrants a note on the estimation of epistemic uncertainty estimation in general. Firstly, in Evidential Deep Learning, model uncertainty is no longer estimated via the uncertainty in learnable model parameters, but instead through the parameters of the prior or posterior distribution. Furthermore, distinguishing epistemic from distributional uncertainty allows to differentiate uncertainty due to underspecification from uncertainty due to a lack of evidence. In BMA, these notions are indistinguishable: In theory, model uncertainty on OOD should be high since the model is underspecified on them, however empirical works have shown this to not always be the case (Ulmer et al., 2020; Ulmer & Cinà, 2021; Van Landeghem et al., 2022). In both cases, it is impossible to estimate the uncertainty induced by a misspecified model class, which is why approaches using auxiliary models directly predicting the model uncertainty have been proposed (Jain et al., 2021).

### 3.3 Existing Approaches for Dirichlet Networks

Being able to quantify aleatoric, epistemic and distributional uncertainty in one forward pass and in closed form are desirable traits, as they simplify the process of obtaining different uncertainty scores. However, it is important to note that the behaviors of the Dirichlet distributions in Figure 2 are idealized. In the empirical risk minimization framework that neural networks are usually trained in, Dirichlet networks are

---

[4]With exceptions such as Haussmann et al., 2019; Zhao et al., 2020). When the distribution over parameters in Equation (5) is retained, alternate expressions of the aleatoric and epistemic uncertainty are derived by (Woo, 2022).

Table 1: Overview over prior networks for classification. ($*$) OOD samples were created inspired by the approach of Liang et al. (2018). ID: Using in-distribution data samples.

| Method | Loss function | Architecture | Requires OOD samples? |
|---|---|---|---|
| Prior network (Malinin & Gales, 2018) | ID KL w.r.t smoothed label & OOD KL w.r.t. uniform prior | MLP / CNN | ✓ |
| Prior networks (Malinin & Gales, 2019) | Reverse KL of Malinin & Gales (2018) | CNN | ✓ |
| Information Robust Dirichlet Networks (Tsiligkaridis, 2019) | $l_p$ norm w.r.t one-hot label & Approx. Rényi divergence w.r.t. uniform prior | CNN | ✗ |
| Dirichlet via Function Decomposition (Biloš et al., 2019) | Uncertainty Cross-entropy & mean & variance regularizer | RNN | ✗ |
| Prior network with PAC Regularization (Haussmann et al., 2019) | Negative log-likelihood loss + PAC regularizer | BNN | ✗ |
| Ensemble Distribution Distillation (Malinin et al., 2020b) | Knowledge distillation objective | MLP / CNN | ✗ |
| Prior networks with representation gap (Nandy et al., 2020) | ID & OOD Cross-entropy + precision regularizer | MLP / CNN | ✓ |
| Prior RNN (Shen et al., 2020) | Cross-entropy + entropy regularizer | RNN | (✓)$^*$ |
| Graph-based Kernel Dirichlet distribution estimation (GKDE) (Zhao et al., 2020) | $l_2$ norm w.r.t. one-hot label & KL reg. with node-level distance prior & Knowledge distillation objective | GNN | ✗ |

not incentivized to behave in the depicted way per se. Thus, when comparing existing approaches for parameterizing Dirichlet priors (Section 3.3.1) and posteriors (Section 3.3.2),[5] we mainly focus on the different ways in which authors try to tackle this problem by means of loss functions and training procedures. We give an overview over the discussed works in Tables 1 and 2 in the respective sections. For additional details, we refer the reader to Appendix B for general derivations concerning the Dirichlet distribution. We dedicate Appendix C to more extensive derivations of the different loss functions and regularizers and give a detailed overview over their mathematical forms in Appendix D.

### 3.3.1 Prior Networks

The key challenge in training Dirichlet networks is to ensure both high classification performance and the intended behavior under foreign data inputs. For this reason, most discussed works follow a loss function design using two parts: One optimizing for task accuracy for the former goal, the other one for a flat Dirichlet distribution for the latter, as flatness suggests a lack of evidence. To enforce flatness, the predicted Dirichlet is compared to the target distribution using some probabilistic divergence measure. We divide prior networks into two groups: Approaches using additional OOD data for this purpose (*OOD-dependent approaches*), and those without that necessity (*OOD-free approaches*).

**OOD-free approaches** Apart from a standard negative log-likelihood loss (NLL) like used in Haussmann et al. (2019), one simple approach in optimizing the model is to impose a $l_p$-loss between the one-hot encoded label $\mathbf{y}$ and the categorical distribution $\boldsymbol{\mu}$. Tsiligkaridis (2019) show that since the values of $\boldsymbol{\mu}$ in turn depend directly on the predicted concentration parameters $\boldsymbol{\alpha}$, the generalized (upper bound to the) loss can be derived to be the following (see the full derivation is given in Appendix C.3):

---

[5]Even though the term *prior* and *posterior network* have been coined by Malinin & Gales (2018) and Charpentier et al. (2020) for their respective approaches, we use them in the following as an umbrella term for all methods targeting a prior or posterior distribution.

$$\mathbb{E}_{p(\boldsymbol{\mu}|\mathbf{x},\boldsymbol{\theta})}\big[||\,\mathbf{y}-\boldsymbol{\mu}||_p\big] \leq \left(\frac{\Gamma(\alpha_0)}{\Gamma(\alpha_0+p)}\right)^{\frac{1}{p}} \left(\frac{\Gamma\big(\sum_{k\neq y}\alpha_k+p\big)}{\Gamma\big(\sum_{k\neq y}\alpha_k\big)} + \sum_{k\neq y}\frac{\Gamma(\alpha_k+p)}{\Gamma(\alpha_k)}\right)^{\frac{1}{p}}$$

Since the sum over concentration parameters excludes the one corresponding to the gold label, this loss can be seen as reducing the density on the areas of the probability simplex that do not correspond to the target class. Zhao et al. (2020) specifically utilize the $l_2$ loss, which has the following form (see Appendix C.4):

$$\mathbb{E}_{p(\boldsymbol{\mu}|\mathbf{x},\boldsymbol{\theta})}\Big[||\,\mathbf{y}-\boldsymbol{\mu}||_2^2\Big] = \sum_{k=1}^{K}\left(\mathbf{1}_{y=k}-\frac{\alpha_k}{\alpha_0}\right)^2 + \frac{\alpha_k(\alpha_0-\alpha_k)}{\alpha_0^2(\alpha_0+1)}$$

where $\mathbf{1}$ denotes the indicator function. Since $\alpha_k/\alpha_0 \leq 1$, we can see that the term with the indicator functions penalizes the network when the concentration parameter $\alpha_k$ corresponding to the correct label does not exceed the others. The remaining aspect lies in the regularization: For reliable predictive uncertainty, the density associated with incorrect classes should be reduced. One such option is to increase the Kullback-Leibler divergence w.r.t. a uniform Dirichlet (see Appendix B.3):

$$\mathrm{KL}\Big[p(\boldsymbol{\mu}|\boldsymbol{\alpha})\big|\big|p(\boldsymbol{\mu}|\mathbf{1})\Big] = \log\frac{\Gamma(K)}{B(\boldsymbol{\alpha})} + \sum_{k=1}^{K}(\alpha_k-1)\big(\psi(\alpha_k)-\psi(\alpha_0)\big)$$

Since Zhao et al. (2020) apply their model to graph structures, they do not increase the divergence to a uniform Dirichlet, but incorporate information about the local graph neighborhood into the reference distribution by considering the distance and label of close nodes.[6] Nevertheless, we will see the KL-divergence w.r.t. a uniform Dirichlet be used by many of the following works. Other divergence measures are also possible: Tsiligkaridis (2019) instead use a local approximation of the Rényi divergence.[7] First, the density for the correct class is removed from the Dirichlet by creating $\tilde{\boldsymbol{\alpha}} = (1-\mathbf{y})\cdot\boldsymbol{\alpha}+\mathbf{y}$. Then, the remaining concentration parameters are encouraged towards uniformity by the divergence

$$\mathrm{R\acute{e}nyi}\Big[p(\boldsymbol{\mu}|\tilde{\boldsymbol{\alpha}})\big|\big|p(\boldsymbol{\mu}|\mathbf{1})\Big] \approx \frac{1}{2}\Big[\sum_{k\neq y}(\alpha_k-1)^2\big(\psi^{(1)}(\alpha_j)-\psi^{(1)}(\tilde{\alpha}_0)\big) - \psi^{(1)}(\tilde{\alpha}_0)\sum_{\substack{k\neq k' \\ k\neq y,\ k'\neq y}}(\alpha_k-1)(\alpha_{k'}-1)\Big]$$

where where $\psi^{(1)}$ denotes the first-order polygamma function, defined as $\psi^{(1)}(x) = \frac{d}{dx}\psi(x)$. Haussmann et al. (2019) derive an entirely different regularizer using Probably Approximately Correct (PAC) bounds from learning theory, that together with the negative log-likelihood gives a proven bound to the expected true risk of the classifier. Setting a scalar $\delta$ allows to set the desired risk, i.e. the model's expected risk is guaranteed to be the same or less than the derived PAC bound with a probability of $1-\delta$. For a problem with $N$ available training data points, the following upper bound for the regularizer is presented:

$$\sqrt{\frac{\mathrm{KL}\big[p(\boldsymbol{\mu}|\boldsymbol{\alpha})\big|\big|p(\boldsymbol{\mu}|\mathbf{1})\big]-\log\delta}{N}}-1$$

We see that even from the learning-theoretic perspective, this method follows the intuition of the original KL regularizer. Haussmann et al. (2019) also admit that in this form, the regularizer does not allow for a

---

[6]They also add another knowledge distillation term (Hinton et al., 2015) to their loss, for which the model tries to imitate the predictions of a vanilla Graph Neural Network that functions as the teacher network.

[7]The Kullback-Leibler divergence can be seen as a special case of the Rényi divergence (van Erven & Harremoës, 2014), where the latter has a stronger information-theoretic underpinning.

direct PAC interpretation anymore, but demonstrate its usefulness in their experiments. Summarizing all of the presented approaches thus far, we can see that they try to force to model to concentrate the Dirichlet's density solely on the parameter corresponding to the right label — expecting a more flat density for difficult or unknown inputs. The latter behavior can also be achieved explicitly by training with OOD inputs in a series of works we discuss next.

**OOD-dependent approaches** Note that before we were maximizing the KL-divergence to a uniform Dirichlet, in order to concentrate all the density in the correct corner of the probability simplex. Now, Malinin & Gales (2018) instead explicitly *minimize* the KL divergence to a uniform Dirichlet on OOD data points. This way, the model is encouraged to be agnostic about its prediction in the face of unknown inputs. Further, they utilize another KL term to train the model on predicting the correct label instead of a $l_p$ norm, minimizing the distance between the predicted concentration parameters and the gold label. However, since only a gold *label* and not a gold *distribution* is available, they create one by re-distributing some of the density from the correct class onto the rest of the simplex (see Appendix D for full form). In their follow-up work, Malinin & Gales (2019) argue that the asymmetry of the KL divergence as the main objective creates unappealing properties in producing the correct behavior of the predicted Dirichlet, since it creates a multi- instead of unimodal target distribution. They therefore propose to use the reverse KL instead (see Appendix C.5 for the derivation). Nandy et al. (2020) refine this idea further, stating that even in with reverse KL training high epistemic and high distributional uncertainty (Figures 2d and 2e) might be confused, and instead propose novel loss functions producing a *representation gap* (Figure 2f), which aims to be more easily distinguishable. In this case, spread out densities signify epistemic uncertainty, whereas densities concentrated entirely on the edges of the simplex indicate distributional uncertainty. The way they achieve this goal is two-fold: In addition to minimizing the NLL loss on in-domain and maximizing the entropy on OOD examples, they also penalize the precision of the Dirichlet (see Appendix D for full form). Maximizing the entropy on OOD examples hereby serves the same function as minimizing the KL w.r.t to a uniform distribution, and can be implemented using the closed-form solution in Appendix B.2:

$$H\big[p(\boldsymbol{\mu}|\,\boldsymbol{\alpha})\big] = \log B(\boldsymbol{\alpha}) + (\alpha_0 - K)\psi(\alpha_0) - \sum_{k=1}^{K}(\alpha_k - 1)\psi(\alpha_k)$$

**Knowledge distillation** A way to forego the use of OOD examples while still using external information for regularization is to use *knowledge distillation* (Hinton et al., 2015). Here, the core idea lies in a student model learning to imitate the predictions of a more complex teacher model. Malinin et al. (2020b) exploit this idea and show that prior networks can also be distilled using an ensemble of classifiers and their predicted categorical distributions (akin to learning Figure 2e from Figure 2a), which does not require regularization at all, but comes at the cost of having to train an entire ensemble.

**Sequential models** We identified two sequential applications of prior networks in the literature: For Natural Language Processing, Shen et al. (2020) train a recurrent neural network for spoken language understanding using a simple cross-entropy loss. Instead of using OOD examples for training, they aim maximize the entropy of the model on data inputs given a learned, noisy version of the predicted concentration parameters. In comparison, Biloš et al. (2019) apply their model to asynchronous event classification and note that the standard cross-entropy loss only involves a point estimate of a categorical distribution, discarding all the information contained in the predicted Dirichlet. For this reason, they propose an *uncertainty-aware* cross-entropy (UCE) loss instead, which has a closed-form solution in the Dirichlet case (see Appendix C.6):

$$\mathcal{L}_{\text{UCE}} = \psi(\alpha_y) - \psi(\alpha_0)$$

Since their final concentration parameters are created using additional information from a class-specific Gaussian process, they further regularize the mean and variance for OOD data points using an extra loss term, incentivizing a loss mean and a variance corresponding to a pre-defined hyperparameter.

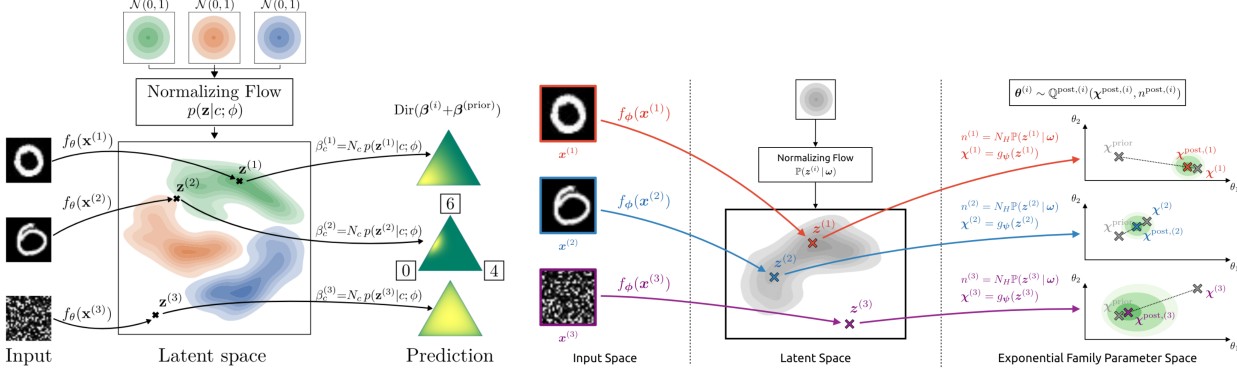

(a) Posterior Network (Charpentier et al., 2020).    (b) Natural Posterior Network (Charpentier et al., 2021).

Figure 3: Schematic of the Posterior Network and Natural Posterior Network, taken from Charpentier et al. (2020; 2021), respectively. In both cases, an encoder $f_\theta$ maps inputs to a latent representation $\mathbf{z}$. NFs then model the latent densities, which are used together with the prior concentration to produce the posterior parameters. For $\mathbf{x}^{(1)}$, its latent representation lies right in the modelled density of the first class, and thus receives a confident prediction. The latent $\mathbf{z}^{(2)}$ lies between densities, creating aleatoric uncertainty. $\mathbf{x}^{(3)}$ is an OOD input, is mapped to a low-density area of the latent space and thus produces an uncertain prediction. The differences in the two approaches is that the Posterior Network in (a) uses one NF per class, while only one NF is used in (b). Furthermore, (b) constitutes a generalization to different exponential family distributions, and is not restricted to classification problems (see main text for more detail).

Table 2: Overview over posterior networks for classification. OOD samples were created via (†) the fast-sign gradient method (Kurakin et al.), using a (‡) Variational Auto-Encoder (VAE; Kingma & Welling, 2014 or a (§) Wasserstein GAN (WGAN; Arjovsky et al., 2017). NLL: Negative log-likelihood. CE: Cross-entropy.

| Method | Loss function | Architecture | Requires OOD samples? |
|---|---|---|---|
| Evidential Deep Learning (Sensoy et al., 2018) | $l_2$ norm w.r.t. one-hot label + KL w.r.t. uniform prior | CNN | ✗ |
| Regularized ENN Zhao et al. (2019) | $l_2$ norm w.r.t. one-hot label + Uncertainty regularizer on OOD/ difficult samples | MLP / CNN | ✓ |
| WGAN–ENN (Hu et al., 2021) | $l_2$ norm w.r.t. one-hot label + Uncertainty regularizer on synth. OOD | MLP / CNN + WGAN | (✓)[§] |
| Variational Dirichlet (Chen et al., 2018) | ELBO + Contrastive Adversarial Loss | CNN | (✓)[†] |
| Belief Matching (Joo et al., 2020) | ELBO | CNN | ✗ |
| Posterior Networks (Charpentier et al., 2020) | Uncertainty CE (Biloš et al., 2019) + Entropy regularizer | MLP / CNN + Norm. Flow | ✗ |
| Graph Posterior Networks (Stadler et al., 2021) | Same as Charpentier et al. (2020) | GNN | ✗ |
| Generative Evidential Neural Networks (Sensoy et al., 2020) | Contrastive NLL + KL between uniform & Dirichlet of wrong classes | CNN | (✓)[‡] |

### 3.3.2 Posterior Networks

As elaborated on in Section 2.1, choosing a Dirichlet prior, due to its conjugacy to the categorical distribution, induces a Dirichlet posterior distribution. Like the prior in the previous section, this posterior can be parameterized by a neural network. The challenges hereby are two-fold: Accounting for the number of class observations $N_k$ that make up part of the posterior density parameters $\beta$ (Equation (3)), and, similarly to prior networks, ensuring the wanted behavior on the probability simplex for in- and out-of-distribution

inputs. Sensoy et al. (2018) base their approach on the Dempster-Shafer theory of evidence (Yager & Liu, 2008; lending its name to the term "Evidential Deep Learning") and its formalization via subjective logic (Audun, 2018). In doing so, an agnostic belief in form of a uniform Dirichlet prior $\forall k : \alpha_k = 1$ is updated using pseudo-counts $N_k$, which are predicted by a neural network. This is different from prior networks, where the concentration parameters are predicted instead. In both cases, this does not require any modification to a model's architecture except for replacing the softmax output function by a ReLU (or similar). Sensoy et al. (2018) train their model using the same techniques presented in the previous section: The main objective is the $l_2$ loss, penalizing the difference between the predicted Dirichlet and the one-hot encoded class label (Appendix C.4), and the KL divergence w.r.t. a uniform Dirichlet is used for regularization.

**Generating OOD samples using generative models** Since OOD examples are not always readily available, several works try to create artificial samples using deep generative models. Hu et al. (2021) train a Wasserstein GAN (Arjovsky et al., 2017) to generate OOD samples, on which the network's uncertainty is maximized. The uncertainty is given through *vacuity*, defined as $K/\sum_k \beta_k$. In a follow-up work, Sensoy et al. (2020) similarly train a model using a contrastive loss with artificial OOD samples from a Variational Autoencoder (Kingma & Welling, 2014), and a KL-based regularizer similar to that of Tsiligkaridis (2019), where the density for posterior concentration parameters $\beta_k$ that do not correspond to the gold label are pushed to the uniform distribution.

**Posterior networks via Normalizing Flows** Charpentier et al. (2020) also set $\boldsymbol{\alpha}$ to a uniform prior, but obtain the pseudo-observations $N_k$ in a different way: Instead of a model predicting them directly, $N_k$ is determined by the number of examples of a certain class in the training set. This quantity is further modified in the following way: An encoder model $f_{\boldsymbol{\theta}}$ produces a latent representation $\mathbf{z}$ of some input. A (class-specific) normalizing flow[8] (NF; Rezende & Mohamed, 2015) then assigns a probability to this latent representation, which is used to weight $N_k$:

$$\beta_k = \alpha_k + N_k \cdot p(\mathbf{z}\,|\,y = k, \boldsymbol{\phi}); \quad \mathbf{z} = f_{\boldsymbol{\theta}}(\mathbf{x})$$

This has the advantage of producing low probabilities for strange inputs like the noise as depicted in Figure 3a, which in turn translate to low concentration parameters of the posterior Dirichlet, as it falls back onto the uniform prior. The model is optimized using the same uncertainty-aware cross-entropy loss as in Biloš et al. (2019) with an additional entropy regularizer, encouraging density only around the correct class. This scheme is also applied to Graph Neural Networks by Stadler et al. (2021): In order to take the neighborhood structure of the graph into account, the authors also use a Personalized Page Rank scheme to diffuse node-specific posterior parameters $\boldsymbol{\beta}$ between neighboring nodes. The Page Rank scores, reflecting the importance of a neighboring node to the current node, can be approximated using power iteration (Klicpera et al., 2019) and used to aggregate the originally predicted concentration parameters $\boldsymbol{\beta}$ on a per-node basis.

Charpentier et al. (2021) generalize the posterior network approach for exponential family distributions. Akin to the update for the posterior Dirichlet parameters, they formulate a general Bayesian update rule as

$$\boldsymbol{\chi}_i^{\text{post}} = \frac{n^{\text{prior}}\boldsymbol{\chi}^{\text{prior}} + n_i\,\boldsymbol{\chi}_i}{n^{\text{prior}} + n_i}; \quad \mathbf{z}_i = f_{\boldsymbol{\theta}}(\mathbf{x}_i); \quad n_i = N \cdot p(\mathbf{z}\,|\,\boldsymbol{\phi}); \quad \boldsymbol{\chi}_i = g_{\boldsymbol{\psi}}(\mathbf{x}_i) \tag{10}$$

$\boldsymbol{\chi}$ here denotes the distributional parameters and $n$ the evidence. Thus, posterior parameters for a sample $\mathbf{x}_i$ are obtained by updating the prior parameter and some prior evidence by some input-dependent pseudo-evidence $n_i, \boldsymbol{\chi}_i$: Again, given a latent representation by an encoder $\mathbf{z}$, a (this time single) normalizing flow predicted $n_i = N_H \cdot p(\mathbf{z}\,|\,\boldsymbol{\phi})$ based on some pre-defined certainty budget $N_H$,[9] and parameters $\boldsymbol{\chi}_i$ are predicted

---

[8]A NF is a generative model, estimating a density in the feature space by mapping it to a Gaussian in a latent space by a series of invertible, bijective transformations. The probability of an input can then be estimated by calculating the probability of its latent encoding under that Gaussian and applying the change-of-variable formula, traversing the flow in reverse. Instead of mapping from the feature space into latent space, the flows in Charpentier et al. (2020) map from the encoder latent space into a separate, second latent space.

[9]The certainty budget can simply be set to the number of available datapoints, however Charpentier et al. (2021) suggest to set it to $\log N_H = \frac{1}{2}\left(H\log(2\pi) + \log(H+1)\right)$ to better scale with the dimensionality of the latent space $H$.

Table 3: Overview over Evidential Deep Learning methods for regression.

| Method | Parameterized distribution | Loss function | Model |
|---|---|---|---|
| Deep Evidential Regression (Amini et al., 2020) | Normal-Inverse Gamma Prior | Negative log-likelihood loss + KL w.r.t. uniform prior | MLP / CNN |
| Deep Evidential Regression with Multi-task Learning (Oh & Shin, 2021) | Normal-Inverse Gamma Prior | Like Amini et al. (2020), with additional Lipschitz-modified MSE loss | MLP / CNN |
| Multivariate Deep Evidential Regression (Meinert & Lavin, 2021) | Normal-Inverse Wishart Prior | Like Amini et al. (2020), but tying two predicted params. instead of using a regularizer | MLP |
| Regression Prior Network (Malinin et al., 2020a) | Normal-Wishart Prior | Reverse KL (Malinin & Gales, 2019) | MLP / CNN |
| Natural Posterior Network (Charpentier et al., 2021) | Inverse-$\chi^2$ Posterior | Uncertainty Cross-entropy (Biloš et al., 2019) + Entropy regularizer | MLP / CNN + Norm. Flow |

by an additional network $\boldsymbol{\chi}_i = g_{\boldsymbol{\psi}}(\mathbf{z})$, see Figure 3b. For classification, $n^{\text{prior}} = 1$ and $\boldsymbol{\chi}^{\text{prior}}$ corresponds to the uniform Dirichlet, while $\boldsymbol{\chi}_i$ are concentration parameters predicted by an output layer based on the input's latent encoding. We will discuss the same method applied to regression in the next section.

**Posterior networks via variational inference**   Another route lies in directly parameterizing the posterior parameters $\boldsymbol{\beta}$. Given a target distribution defined by a uniform Dirichlet prior plus the number of times an input is associated with a specific label, Chen et al. (2018) optimize a distribution matching objective, i.e. the KL-divergence between the posterior parameters predicted by a neural network and the target distribution. Since this objective is intractable to optimize directly, this leaves us to instead model an *approximate posterior* using variational inference methods, which is exactly the approach of Joo et al. (2020) and Chen et al. (2018). As the KL divergence between the true and approximate posterior is infeasible to estimate, the variational methods usually optimizes the *evidence lower bound* (ELBO) instead:

$$\mathcal{L}_{\text{ELBO}} = \underbrace{\psi(\beta_y) - \psi(\beta_0)}_{\text{UCE loss}} \underbrace{- \log \frac{B(\boldsymbol{\beta})}{B(\boldsymbol{\gamma})} + \sum_{k=1}^{K} (\beta_k - \gamma_k)\big(\psi(\beta_k) - \psi(\beta_0)\big)}_{\text{KL-divergence}}$$

in which we can identify to consist of the uncertainty-aware cross-entropy loss used by Biloš et al. (2019); Charpentier et al. (2020; 2021) and the KL-divergence between two Dirichlets (Appendix B.3).

## 4   Evidential Deep Learning for Regression

Because the Evidential Deep Learning framework provides convenient uncertainty estimation, the question naturally arises of whether it can be extended to regression problems as well. The answer is affirmative, although the Dirichlet distribution is not an appropriate choice in this case. It is very common to model a regression problem using a normal likelihood (Bishop, 2006). As such, there are multiple potential choices for a prior distribution. The methods listed in Table 3 either choose the Normal-Inverse Gamma distribution (Amini et al., 2020; Charpentier et al., 2021), inducing a scaled inverse-$\chi^2$ posterior (Gelman et al., 1995),[10] or as a Normal-Wishart prior (Malinin et al., 2020a). We will discuss these approaches in turn.

**Univariate regression**   Amini et al. (2020) model the regression problem as a normal distribution with unknown mean and variance $\mathcal{N}(y; \mu, \sigma^2)$, and use a normal prior for the mean with $\mu \sim \mathcal{N}(\gamma, \sigma^2 \nu^{-1})$ and an inverse Gamma prior for the variance with $\sigma^2 \sim \Gamma^{-1}(\alpha, \beta)$, resulting in a combined Inverse-Gamma prior with parameters $\gamma, \nu, \alpha, \beta$, shown in Figure 4. These are predicted by different "heads" of a neural network.

---

[10]The form of the Normal-Inverse Gamma posterior and the Normal Inverse-$\chi^2$ posterior are interchangable using some parameter substitutions (Murphy, 2007).

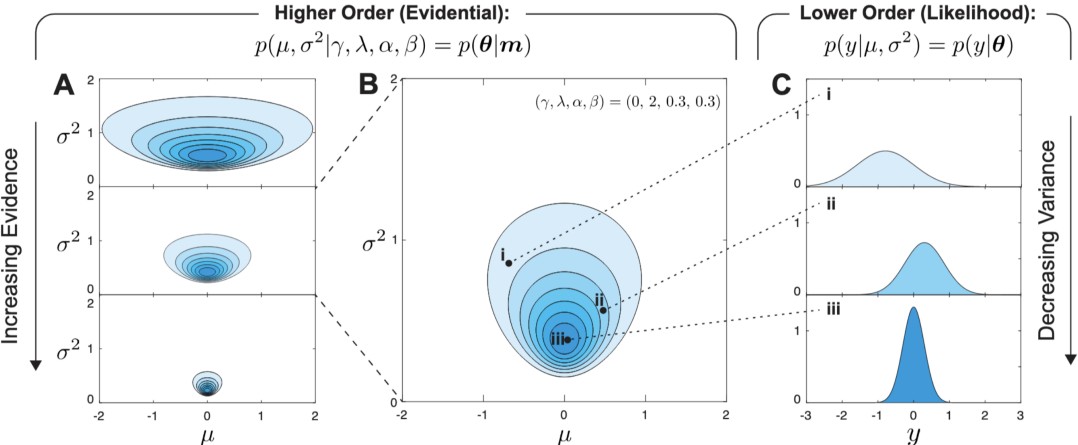

Figure 4: Example of an application of Evidential Deep Learning for regression, taken from Amini et al. (2020). The neural network predicts an Normal Inverse-Gamma prior, whose corresponding normal likelihoods display decreasing variance (and thus uncertainty) in the face of stronger evidence.

For predictions, the expectation of the mean $\mathbb{E}[\mu] = \gamma$, and aleatoric and epistemic uncertainty can then be estimated using the expected value of the variance as well as the variance of the mean, respectively, which have closed form solutions under this parameterization:

$$\mathbb{E}[\sigma^2] = \frac{\beta}{\alpha - 1}; \quad \mathrm{Var}[\mu] = \frac{\beta}{\nu(\alpha - 1)}$$

The model is optimized using a negative log-likelihood objective, which can be analytically be shown to correspond to a Student's t-distribution with $\gamma$ degrees of freedom, mean $\beta(1 + \nu)/(\nu\alpha)$ and variance $2\alpha$:

$$\mathcal{L}_{\mathrm{NLL}} = \frac{1}{2} \log\left(\frac{\pi}{\nu}\right) - \alpha \log \Omega + \left(\alpha + \frac{1}{2}\right) \log\left((y_i - \gamma)^2 \nu + \Omega\right) + \log\left(\frac{\Gamma(\alpha)}{\Gamma(\alpha + \frac{1}{2})}\right) \tag{11}$$

using $\Omega = 2\beta(1 + \nu)$. Akin to the entropy regularizer for Dirichlet networks, Amini et al. (2020) propose a regularization term that concentrates density on the correct prediction:

$$\mathcal{L}_{\mathrm{reg}} = |y_i - \gamma| \cdot (2\nu + \alpha) \tag{12}$$

Since $\mathbb{E}[\mu] = \gamma$ is the prediction of the network, the second term in the product will be scaled by the degree to which the current prediction deviates from the target value. Since $\nu$ and $\alpha$ control the variances of the the mean and variances of the normal likelihood, this term encourages the network to decrease the evidence for mispredicted data samples. As Amini et al. (2020) point out, large amounts of evidence are not punished in cases where the prediction is close to the target. However, Oh & Shin (2021) argue that this combination of objective might create adverse incentives for the model during training: Since the difference between the prediction and target in Equation (11) is scaled by $\nu$, the model could learn to increase the predictive uncertainty by decreasing $\nu$ instead of improving its prediction. They propose to ameliorate this issue by using a third loss term of the form

$$\mathcal{L}_{\mathrm{MSE}} = \begin{cases} (y_i - \gamma)^2 & \text{if } (y_i - \gamma)^2 < U_{\nu,\alpha} \\ 2\sqrt{U_{\nu,\alpha}}|y_i - \gamma| - U_{\nu,\alpha} & \text{if } (y_i - \gamma)^2 \geq U_{\nu,\alpha} \end{cases} \tag{13}$$

where $U_{\nu,\alpha}$ denotes the minimum value for the uncertainty thresholds for $\nu, \alpha$ given over a mini-batch, which are themselves defined as

$$U_\nu = \frac{\beta(\nu+1)}{\alpha\nu}; \quad U_\alpha = \frac{2\beta(\nu+1)}{\nu}\left[\exp\left(\psi(\alpha+\frac{1}{2}) - \psi(\alpha)) - 1\right)\right]. \tag{14}$$

These expression are obtained by taking the derivatives $\partial\mathcal{L}_{\mathrm{NLL}}/\partial\nu$, $\partial\mathcal{L}_{\mathrm{NLL}}/\partial\alpha$ and solving for the parameters, thus giving us the values for $\nu$ and $\alpha$ for which the loss gradients are maximal. In combination with Equation (13), Equation (14) ensures that should the model error exceed $U_{\nu,\alpha}$, the error is rescaled and thus bounds the Lipschitz constant of the loss function, motivating the model to ensure the correctness of its prediction instead of increasing its uncertainty.

**Posterior networks for regression** Another approach for regression is the Natural Posterior Network by Charpentier et al. (2021), which was already discussed for classification in Section 3.3.2. But since the proposed approach is a generalization for exponential family distributions, it can be applied to regression as well, using a Normal likelihood and Normal Inverse-Gamma prior as well. The Bayesian update rule in Equation (10) is adapted as follows: $n$ is set to $n = \lambda = 2\alpha$, and $\boldsymbol{\chi} = \left[\mu_0 \mid \mu_0^2 + 2\beta/n\right]^T$. Feeding an input into the natural posterior network again first produces a latent encoding $\mathbf{z}$, from which a NF predicts $n_i = N_H \cdot p(\mathbf{z}\,|\,\boldsymbol{\phi})$, and an additional network produces $\boldsymbol{\chi}_i = g_{\boldsymbol{\psi}}(\mathbf{z})$, which are used in Equation (10) to produce $\boldsymbol{\chi}^{\mathrm{post}}$ and $n^{\mathrm{post}}$, from which the parameters of the posterior Normal Inverse-Gamma can be derived. The authors also produce a general exponential family form of the UCE loss by Biloš et al. (2019), consisting of expected log-likelihood and an entropy regularizer, which they derive for the regression parameterization. Again, this approach relies on the density estimation capabilities of the NF to produce an agnostic belief about the right prediction for OOD examples (see Figure 3b).

**Multivariate evidential regression** There are also some works offering solutions for multivariate regression problems: Malinin et al. (2020a) can be seen as another multivariate generalization of the work of Amini et al. (2020), where a combined Normal-Wishart prior is formed to fit the now Multivariate Normal likelihood. Again, the prior parameters are the output of a neural network, and uncertainty can be quantified in a similar way. For training purposes, they apply two different training objectives using the equivalent of the reverse KL objective of Malinin & Gales (2019) as well as of the knowledge distillation objective of Malinin et al. (2020b), which does not require OOD data for regularization purposes. Meinert & Lavin (2021) also provide a solution using a Normal Inverse-Wishart prior. In a similar vein to Oh & Shin (2021), they argue that the original objective proposed by Amini et al. (2020) can be minimized by increasing the network's uncertainty instead of the decreasing the mismatch of its prediction. As a solution, they simply propose to tie $\beta$ and $\nu$ via a hyperparameter.

# 5 Related Approaches & Applications

**Existing approaches** The need for the quantification of uncertainty in order to earn the trust of end-users and stakeholders has been a key driver for research (Bhatt et al., 2021). Unfortunately, standard neural discriminator architectures have been proven to possess unwanted theoretical properties w.r.t. OOD inputs[11] (Hein et al., 2019; Ulmer & Cinà, 2021) and lacking calibration in practice (Guo et al., 2017). A popular way to overcome these blemishes is by quantifying (epistemic) uncertainty by aggregating multiple predictions by networks in the Bayesian model averaging framework (Jeffreys, 1998; Wilson & Izmailov, 2020; Kristiadi et al., 2020; Daxberger et al., 2021; Gal & Ghahramani, 2016; Blundell et al., 2015; Lakshminarayanan et al., 2017). Nevertheless, many of these methods have been shown not to produce diverse predictions (Wilson & Izmailov, 2020; Fort et al., 2019) and to deliver subpar performance and potentially misleading uncertainty estimates under distributional shift (Ovadia et al., 2019; Masegosa, 2020; Wenzel et al., 2020; Izmailov et al., 2021a;b), raising doubts about their efficacy.

**Related Approaches** Kull et al. (2019) found an appealing use of the Dirichlet distribution as a post-training calibration map. The proposed Posterior Network (Charpentier et al., 2020; 2021) can furthermore be seen as related to another, competing approach, namely the combination of neural discriminators with

---

[11]Pearce et al. (2021) argue that some insights might partially be mislead by low-dimensional intuitions, and that empirically OOD data in higher dimensions tend to be mapped into regions of higher uncertainty.

density estimation methods, for instance in the form of energy-based models (Grathwohl et al.; Elflein et al., 2021) or other hybrid architectures (Lee et al., 2018; Mukhoti et al., 2021).

**Applications**  Some of the discussed models have already found a variety of applications, such as in autonomous driving (Capellier et al., 2019; Liu et al., 2021; Petek et al., 2021; Wang et al., 2021), medical screening (Ghesu et al., 2019; Gu et al., 2021), molecular analysis (Soleimany et al., 2021), open set recognition (Bao et al., 2021), active learning (Hemmer et al., 2022) and model selection (Radev et al., 2021).

## 6  Discussion

**Challenges**  Despite their advantages, the last chapters have highlighted key weaknesses of Dirichlet networks as well: In order to achieve the right behavior of the distribution and thus guarantee sensible uncertainty estimates (since ground truth estimates are not available), the surveyed literature proposes a variety of loss functions. Bengs et al. (2022) show formally that many of the loss functions used so far are *not* appropriate and violate basic asymptotic assumptions about epistemic uncertainty. Furthermore, some approaches Malinin & Gales (2018; 2019); Nandy et al. (2020); Malinin et al. (2020a) require out-of-distribution data points during training. This comes with two problems: Such data is often not available or in the first place, or cannot guarantee robustness against *other* kinds of unseen OOD data, of which infinite types exist in a real-valued feature space.[12]  Indeed, Kopetzki et al. (2021) found OOD detection to deteriorate across a family of Dirichlet-based models under adversarial perturbation and OOD data points.

**Comparison to existing approaches**  As discussed in Section 5, several existing approaches to uncertainty quantification equally suffer from shortcomings with respect to their reliability. One possible explanation for this behavior might lie in the insight that neural networks trained in the empirical risk minimization framework tend to learn spurious but highly predictive features (Ilyas et al., 2019; Nagarajan et al., 2021). This way, inputs stemming from the training distribution can be mapped to similar parts of the latent space as data points outside the distribution even though they have (from a human perspective) blatant semantic differences, simply because these semantic features were not useful to optimize for the training objective. This can result in ID and OOD points having assigned similar feature representations by a network, a phenomenon has been coined "feature collapse" (Nalisnick et al., 2019; van Amersfoort et al., 2021; Havtorn et al., 2021). One strategy to mitigate (but not solve) this issue has been to enforce a constraint on the smoothness of the neural network function (Wei et al., 2018; van Amersfoort et al., 2020; 2021; Liu et al., 2020), thereby maintaining both a sensitivity to semantic changes in the input and robustness against adversarial inputs (Yu et al., 2019). Another approach lies in the usage of OOD data as well, sometimes dubbed "outlier exposure" (Fort et al., 2021), but displaying the same shortcomings as in the EDL case. A generally promising strategy seems to seek functional diversity through ensembling: Juneja et al. (2022) show how model instances ending up in different low-loss modes correspond to distinxt generalization strategies on natural language data, indicating that combining diverse strategies may lead to better generalization and thus potentially also more reliable uncertainty. Attaining different solutions still creates computational overhead, despite new methods to reduce it (Garipov et al., 2018; Dusenberry et al., 2020; Benton et al., 2021).

**Bayesian model averaging**  One of the most fundamental differences between EDL and existing approaches is the sacrifice of Bayesian model averaging (Equations (4) and (5)): In principle, combining multiple parameter estimates in supposed in a lower predictive risk (Fragoso et al., 2018). The Machine Learning community has ascribed further desiderata to this approach, such as better generalization and robustness to distributional shifts. Recent studies with exact Bayesian Neural Networks however have cast doubts on these assumptions (Izmailov et al., 2021a;b). Nevertheless, ensembles, that approximate Equation (4) via Monte Carlo estimates, remain state-of-the-art on many uncertainty benchmarks. EDL abandons modelling epistemic uncertainty through the learnable parameters, and instead expresses it through the uncertainty in prior / posterior parameters. This loses functional diversity which could aid generalization, while sidestepping computational costs. Future research could therefore explore the combination of both paradigms, as proposed by Haussmann et al. (2019); Zhao et al. (2020); Charpentier et al. (2021).

---

[12]The same applies to the artificial OOD data in Chen et al. (2018); Shen et al. (2020); Sensoy et al. (2020).

## 7 Conclusion

This survey has given an overview over contemporary approaches for uncertainty estimation using neural networks to parameterize conjugate priors or the corresponding posteriors instead of likelihoods, called Evidential Deep Learning. We highlighted their appealing theoretical properties allowing for uncertainty estimation with minimal computational overhead, rendering them as a viable alternative to existing strategies. We also emphasized practical problems: In order to nudge models towards the desired behavior in the face of unseen or out-of-distribution samples, the design of the model architecture and loss function have to be carefully considered. Based on a summary and discussion of experimental findings in Appendix A, the entropy regularizer seems to be a sensible choice in prior networks when OOD data is not available. Combining discriminators with generative models like normalizing flows like in (Charpentier et al., 2020; 2021), embedded in a sturdy Bayesian framework, also appears as an exciting direction for practical applications. In summary, we believe that recent advances show promising results for Evidential Deep Learning, making it a viable option in uncertainty estimation to improve safety and trustworthiness in Machine Learning systems.

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

# A   Datasets & Evaluation Techniques Appendix

Table 4: Overview over uncertainty evaluation techniques and datasets. $^{(*)}$ indicates that a dataset was used as an OOD dataset for evaluation purposes, while $^{(\diamond)}$ signifies that it was used as an in-distribution or out-of-distribution dataset. $^{(\dagger)}$ means that a dataset was modified to create ID and OOD splits (for instance by removing some classes for evaluation or corrupting samples with noise).

| | | Data Modality | | |
| --- | --- | --- | --- | --- |
| Method | Uncertainty Evaluation | Images | Tabular | Other |
| Prior network (Malinin & Gales, 2018) | OOD Detection, Misclassification Detection | MNIST, CIFAR-10, Omniglot$^{(*)}$, SVHN$^{(*)}$, LSUN$^{(*)}$, TIM$^{(*)}$ | ✗ | Clusters (Synthetic) |
| Prior networks (Malinin & Gales, 2019) | OOD Detection, Adversarial Attack Detection | MNIST, CIFAR-10/100, SVHN$^{(*)}$, LSUN$^{(*)}$, TIM$^{(*)}$ | ✗ | Clusters (Synthetic) |
| Information Robust Dirichlet Networks (Tsiligkaridis, 2019) | OOD Detection, Adversarial Attack Detection | MNIST, FashionMNIST$^{(*)}$ notMNIST$^{(*)}$, Omniglot$^{(*)}$ CIFAR-10, TIM$^{(*)}$, SVHN$^{(*)}$ | ✗ | ✗ |
| Dirichlet via Function Decomposition (Biloš et al., 2019) | OOD Detection | ✗ | Erdős-Rényi Graph (Synthetic), Stack Exchange, Smart Home, Car Indicators | ✗ |
| Prior network with PAC Regularization (Haussmann et al., 2019) | OOD Detection | MNIST, FashionMNIST$^{(*)}$ CIFAR-10$^{(\dagger)}$ | ✗ | ✗ |
| Ensemble Distribution Distillation (Malinin et al., 2020b) | OOD Detection, Misclassification Detection, Calibration | CIFAR-10, CIFAR-100$^{(\diamond)}$ TIM$^{(\diamond)}$, LSUN$^{(*)}$ | ✗ | Spirals (Synthetic) |
| Prior networks with representation gap (Nandy et al., 2020) | OOD Detection | CIFAR-10$^{(\diamond)}$, CIFAR-100$^{(\diamond)}$ TIM, ImageNet$^{(*)}$ | ✗ | Clusters (Synthetic) |
| Prior RNN (Shen et al., 2020) | New Concept Extraction | ✗ | ✗ | Concept Learning$^{(\diamond)}$, Snips$^{(\diamond)}$, ATIS$^{(\diamond)}$ (Language) |
| Graph-based Kernel Dirichlet distribution estimation (GKDE) (Zhao et al., 2020) | OOD Detection Misclassification Detection | ✗ | ✗ | Coauthors Physics$^{(\diamond)}$, Amazon Computer$^{(\diamond)}$ Amazon Photo$^{(\diamond)}$ (Graph) |
| Evidential Deep Learning (Sensoy et al., 2018) | OOD Detection, Adversarial Attack Detection | MNIST, notMNIST$^{(*)}$, CIFAR-10$^{(\dagger)}$ | ✗ | ✗ |
| Regularized ENN Zhao et al. (2019) | OOD Detection | CIFAR-10$^{(\dagger)}$ | ✗ | Clusters (Synthetic) |
| WGAN–ENN (Hu et al., 2021) | OOD Detection, Adversarial Attack Detection | MNIST, notMNIST$^{(*)}$, CIFAR-10$^{(\dagger)}$ | ✗ | Clusters (Synthetic) |
| Variational Dirichlet (Chen et al., 2018) | OOD Detection, Adversarial Attack Detection | MNIST, CIFAR-10/100, iSUN$^{(*)}$, LSUN$^{(*)}$, SVHN$^{(*)}$, TIM$^{(*)}$ | ✗ | ✗ |
| Belief Matching (Joo et al., 2020) | OOD Detection, Calibration | CIFAR-10/100, SVHN$^{(*)}$ | ✗ | ✗ |
| Posterior Networks (Charpentier et al., 2020) | OOD Detection, Misclassification Detection, Calibration | MNIST, FashionMNIST$^{(*)}$, K-MNIST$^{(*)}$, CIFAR-10, SVHN$^{(*)}$ | Segment$^{(\dagger)}$, Sensorless Drive$^{(\dagger)}$ | Clusters (Synthetic) |
| Graph Posterior Networks (Stadler et al., 2021) | OOD Detection, Misclassification Detection, Calibration | ✗ | ✗ | Amazon Computer$^{(\diamond)}$, Amazon Photo$^{(\diamond)}$ CoraML$^{(\diamond)}$, CiteSeerCoraML$^{(\diamond)}$, PubMed$^{(\diamond)}$, Coauthors Physics$^{(\diamond)}$, CoauthorsCS$^{(\diamond)}$, OBGN Arxiv$^{(\diamond)}$ (Graph) |
| Deep Evidential Regression (Amini et al., 2020) | OOD Detection, Misclassification Detection, Adversarial Attack Detection Calibration | NYU Depth v2 ApolloScape* (Depth Estimation) | UCI Regression Benchmark | Univariate Regression (Synthetic) |
| Deep Evidential Regression with Multi-task Learning (Oh & Shin, 2021) | OOD Detection, Calibration | ✗ | Davis, Kiba$^{(\dagger)}$, BindingDB, PubChem$^{(*)}$ (Drug discovery), UCI Regression Benchmark | Univariate Regression (Synthetic) |
| Multivariate Deep Evidential Regression Meinert & Lavin (2021) | Qualitative Evaluation | ✗ | ✗ | Multivariate Regression (Synthetic) |
| Regression Prior Network (Malinin et al., 2020a) | OOD Detection | NYU Depth v2°, KITTI° (Depth Estimation) | UCI Regression Benchmark | Univariate Regression (Synthetic) |
| Natural Posterior Network (Charpentier et al., 2021) | OOD Detection, Calibration | NYU Depth v2, KITTI*, LSUN$^{(*)}$ (Depth Estimation), MNIST, FashionMNIST$^{(*)}$, K-MNIST$^{(*)}$, CIFAR-10$^{(\dagger)}$, SVHN$^{(*)}$, CelebA$^{(*)}$ | UCI Regression Benchmark$^{(\dagger)}$, Sensorless Drive$^{(\dagger)}$, Bike Sharing$^{(\dagger)}$ | Clusters (Synthetic), Univariate Regression (Synthetic)) |

This section contains a discussion of the used datasets, methods to evaluate the quality of uncertainty evaluation, as well as a direct of available models based on the reported results to determine the most useful choices for practitioners. An overview over the differences between the surveyed works is given in Table 4.

**Datasets** Most models are applied to image classification problems, where popular choices involve the MNIST dataset (LeCun, 1998), using as OOD datasets Fashion-MNIST (Xiao et al., 2017), notMNIST (Bulatov, 2011) containing English letters, K-MNIST (Clanuwat et al., 2018) with ancient Japanese Kuzushiji characters, and the Omniglot dataset (Lake et al., 2015), featuring handwritten characters from more than 50 alphabets. Other choices involve different versions of the CIFAR-10 object recognition dataset (LeCun et al., 1998; Krizhevsky et al., 2009) for training purposes and SVHN (Goodfellow et al., 2014), iSUN (Xiao et al., 2010), LSUN (Yu et al., 2015), CelebA (Liu et al., 2015), ImageNet (Deng et al., 2009) and TinyImagenet (Bastidas, 2017) for OOD samples. Regression image datasets include for instance the NYU Depth Estimation v2 dataset (Silberman et al., 2012), using ApolloScape (Huang et al., 2018) or KITTI (Menze & Geiger, 2015) as an OOD dataset. Many authors also illustrate model uncertainty on synthetic data, for instance by simulating clusters of data points using Gaussians (Malinin & Gales, 2018; 2019; Nandy et al., 2020; Zhao et al., 2019; Hu et al., 2020; Charpentier et al., 2020; 2021), spiral data (Malinin et al., 2020b) or polynomials for regression (Amini et al., 2020; Oh & Shin, 2021; Meinert & Lavin, 2021; Malinin et al., 2020a; Charpentier et al., 2021). Tabular datasets include the Segment dataset, predicting image segments based on pixel features (Dua et al., 2017), and the sensorless drive dataset (Dua et al., 2017; Paschke et al., 2013), describing the maintenance state of electric current drives as well as popular regression datasets included in the UCI regression benchmark used by Hernández-Lobato & Adams (2015); Gal & Ghahramani (2016): Boston house prices (Harrison Jr & Rubinfeld, 1978), concrete compression strength (Yeh, 1998), energy efficiency of buildings (Tsanas & Xifara, 2012), forward kinematics of an eight link robot arm (Corke, 1996), maintenance of naval propulsion systems (Coraddu et al., 2016), properties of protein tertiary stuctures, wine quality (Cortez et al., 2009), and yacht hydrodynamics (Gerritsma et al., 1981). Furthermore, Oh & Shin (2021) use a number of drug discovery datasets, such as Davis (Davis et al., 2011), Kiba (Tang et al., 2014), BindingDB (Liu et al., 2007) and PubChem (Kim et al., 2019). Biloš et al. (2019) are the only authors working on asynchronous time even prediction, and supply their own data in the form of processed stack exchange postings, smart home data, and car indicators. Shen et al. (2020) provide the sole method on language data, and use three different concept learning datasets, i.e. Concept Learning (Jia et al., 2017), Snips (Coucke et al., 2018) and ATIS (Hemphill et al., 1990), which contains new OOD concepts to be learned by design. For graph neural networks, Zhao et al. (2020) and Stadler et al. (2021) select data from the co-purchase datasets Amazon Computer, Amazon Photos (McAuley et al., 2015) and the CoraML (McCallum et al., 2000), CiteSeer (Giles et al., 1998) and PubMed (Namata et al., 2012), Coauthors Physics (Shchur et al., 2018), CoauthorCS (Namata et al., 2012) and OGBN Arxiv (Hu et al., 2020) citation datasets. Lastly, Charpentier et al. (2021) use a single count prediction dataset concerned with predicting the number of bike rentals (Fanaee-T & Gama, 2014).

**Uncertainty Evaluation Methods** There usually are no gold labels for uncertainty estimates, which is why the efficacy of proposed solutions has to be evaluated in a different way. One such way used by almost all the surveyed works is using uncertainty estimates in a proxy OOD detection task: Since the model is underspecified on unseen samples from another distribution, it should be more uncertain. By labelling OOD samples as the positive and ID inputs as the negative class, we can measure the performance of uncertainty estimates using the area under the receiver-operator characteristic (AUROC) or the area under the precision-recall curve (AUPR). We can thereby characterize the usage of data from another dataset as a form of covariate shift, while using left-out classes for testing can be seen as a kind of concept shift (Moreno-Torres et al., 2012). Instead of using OOD data, another approach is to use adversarial examples (Malinin & Gales, 2019; Tsiligkaridis, 2019; Sensoy et al., 2018; Hu et al., 2021; Chen et al., 2018; Amini et al., 2020), checking if they can be identified through uncertainty. In the case of Shen et al. (2020), OOD detection or new concept extraction is the actual and not a proxy task, and thus can be evaluated using classical metrics such as the $F_1$ score. Another way is misclassification detection: In general, we would desire the model to be more uncertain about inputs it incurs a higher loss on, i.e., what it is more wrong about. For this purpose, some works (Malinin & Gales, 2018; Zhao et al., 2020; Charpentier et al., 2020) measure

whether let missclassified inputs be the positive class in another binary proxy classification test, and again measure AUROC and AUPR. Alternatively, Malinin et al. (2020b); Stadler et al. (2021); Amini et al. (2020) show or measure the area under the prediction / rejection curve, graphing how task performance varies as predictions on increasingly uncertain inputs is suspended. Lastly, some authors look at a model's calibration (Guo et al., 2017): While this does not allow to judge the quality of uncertainty estimates themselves, quantities like the expected calibration error quantify to what extent the output distribution of a classifier corresponds to the true label distribution, and thus whether aleatoric uncertainty is accurately reflected.

**What is state-of-the-art?**   As apparent from Table 4, evaluation methods and datasets can vary tremendously between different research works. This can make it hard to accurately compare different approaches in a fair manner. Nevertheless, we try to draw some conclusion about the state-of-art in this research direction to the best extent possible: For **image classification**, the posterior (Charpentier et al., 2020) and natural posterior network (Charpentier et al., 2021) provide the best results on the tested benchmarks, both in terms of task performance and uncertainty quality. When the training an extra normalizing flow creates too much computational overhead, prior networks (Malinin & Gales, 2018) with the PAC-based regularizer (Haussmann et al., 2019; see Table 5 for final form) or a simple entropy regularizer (Appendix B.2) can be used. In the case of **regression** problems, the natural posterior network (Stadler et al., 2021) performs better or on par with the evidential regression by Amini et al. (2020) or an ensemble Lakshminarayanan et al. (2017) or MC Dropout (Gal & Ghahramani, 2016). For **graph neural networks**, the graph posterior network (Stadler et al., 2021) and a ensemble provide similar performance, but with the former displaying better uncertainty results. Again, this model requires training a NF, so a simpler fallback is provided by evidential regression (Amini et al., 2020) with the improvement by Oh & Shin (2021). For **NLP** and **count prediction**, the works of Shen et al. (2020) and Charpentier et al. (2021) are the only available instances from this model family, respectively. In the latter case, ensembles and the evidential regression framework (Amini et al., 2020) produce a lower root mean-squared error, but worse uncertainty estimates on OOD.

**Caveats**   Stadler et al. (2021) point out that much of the ability of posterior networks stems from the addition of a NF, which have been shown to also sometimes behave unreliably on OOD data (Nalisnick et al., 2019). Although the NFs in posterior networks operate on the latent and not the feature space, they are also restricted to operate on features that the underlying network has learned to recognize. Recent work by Dietterich & Guyer (2022) has hinted at the fact that networks might identify OOD by the absence of known features, and not by the presence of new ones, providing a case in which posterior networks are likely to fail. Such evidence on OOD data and adversarial examples has indeed been identified by a study by Kopetzki et al. (2021). Exposing the model to (artificial) OOD data during training as done in the case of Malinin & Gales (2018; 2019); Nandy et al. (2020); Zhao et al. (2019) might alleviate this issue, but will never guarantee full coverage of all potential OOD cases. Lastly, Bengs et al. (2022) also prove how some of the used loss function used for training evidential models do not fulfill desiderata for epistemic uncertainty, pointing to a more fundamental flaw.

# B  Fundamental Derivations Appendix

This appendix section walks the reader through generalized versions of recurring theoretical results using Dirichlet distributions in a Machine Learning context, such as their expectation in Appendix B.1, their entropy in Appendix B.2 and the Kullback-Leibler divergence between two Dirichlets in Appendix C.3.

## B.1  Expectation of a Dirichlet

Here, we show results for the quantities $\mathbb{E}[\mu_k]$ and $\mathbb{E}[\log \mu_k]$. For the first, we follow the derivation by Miller (2011). Another proof is given by Lin (2016).

$$\mathbb{E}[\mu_k] = \int \cdots \int \mu_k \frac{\Gamma(\alpha_0)}{\prod_{k'=1}^{K} \Gamma(\alpha_k')} \prod_{k'=1}^{K} \mu_{k'}^{\alpha_{k'}-1} d\mu_1 \ldots d\mu_K$$

Moving $\mu_k^{\alpha_k-1}$ out of the product:

$$= \int \cdots \int \frac{\Gamma(\alpha_0)}{\prod_{k'=1}^{K} \Gamma(\alpha_{k'})} \mu_k^{\alpha_k-1+1} \prod_{k' \neq k} \mu_{k'}^{\alpha_{k'}-1} d\mu_1 \ldots d\mu_K \tag{15}$$

For the next step, we define a new set of Dirichlet parameters with $\beta_k = \alpha_k + 1$ and $\forall k' \neq k : \beta_{k'} = \alpha_{k'}$. For those new parameters, $\beta_0 = \sum_k \beta_k = 1 + \alpha_0$. So by virtue of the Gamma function's property that $\Gamma(\beta_0) = \Gamma(\alpha_0 + 1) = \alpha_0 \Gamma(\alpha_0)$, replacing all terms in the normalization factor yields

$$= \int \cdots \int \frac{\alpha_k}{\alpha_0} \frac{\Gamma(\beta_0)}{\prod_{k'=1}^{K} \Gamma(\beta_{k'})} \prod_{k'=1}^{K} \mu_{k'}^{\beta_{k'}-1} d\mu_1 \ldots d\mu_K = \frac{\alpha_k}{\alpha_0}$$

where in the last step we obtain the final result, since the Dirichlet with new parameters $\beta_k$ must nevertheless integrate to 1, and the integrals do not regard $\alpha_k$ or $\alpha_0$. For the expectation $\mathbb{E}[\log \mu_k]$, we first rephrase the Dirichlet distribution in terms of the exponential family (Kupperman, 1964). The exponential family encompasses many commonly-used distributions, such as the normal, exponential, Beta or Poisson, which all follow the form

$$p(\mathbf{x}; \boldsymbol{\eta}) = h(\mathbf{x}) \exp\left(\boldsymbol{\eta}^T u(\mathbf{x}) - A(\boldsymbol{\eta})\right)$$

with *natural parameters* $\boldsymbol{\eta}$, *sufficient statistic* $u(\mathbf{x})$, and *log-partition function* $A(\boldsymbol{\eta})$. For the Dirichlet distribution, Winn (2004) provides the sufficient statistic as $u(\boldsymbol{\mu}) = [\log \boldsymbol{\mu}_1, \ldots, \boldsymbol{\mu}_K]^T$ and the log-partition function

$$A(\boldsymbol{\alpha}) = \sum_{k=1}^{K} \log \Gamma(\alpha_k) - \log \Gamma(\alpha_o) \tag{16}$$

By Mao (2019), we also find that by the moment-generating function that for the sufficient statistic, its expectation can be derived by

$$\mathbb{E}[u(\mathbf{x})_k] = \frac{\partial A(\boldsymbol{\eta})}{\partial \eta_k} \tag{17}$$

Therefore we can evaluate the expected value of $\log \mu_k$ (i.e. the sufficient statistic) by inserting the definition of the log-partition function in Equation (16) into Equation (17):

$$\mathbb{E}[\log \mu_k] = \frac{\partial}{\partial \alpha_k} \sum_{k=1}^{K} \log \Gamma(\alpha_k) - \log \Gamma(\alpha_0) = \psi(\alpha_k) - \psi(\alpha_0) \tag{18}$$

which corresponds precisely to the definition of the digamma function as $\psi(x) = \frac{d}{dx} \log \Gamma(x)$.

## B.2 Entropy of Dirichlet

The following derivation is adapted from Lin (2016), with the result stated in Charpentier et al. (2020) as well.

$$H[\boldsymbol{\mu}] = -\mathbb{E}[\log p(\boldsymbol{\mu}|\boldsymbol{\alpha})]$$

$$= -\mathbb{E}\left[\log\left(\frac{1}{B(\boldsymbol{\alpha})} \prod_{k=1}^{K} \mu_k^{\alpha_k - 1}\right)\right]$$

$$= -\mathbb{E}\left[-\log B(\boldsymbol{\alpha}) + \sum_{k=1}^{K} (\alpha_k - 1) \log \mu_k\right]$$

$$= \log B(\boldsymbol{\alpha}) - \sum_{k=1}^{K} (\alpha_k - 1)\mathbb{E}[\log \mu_k]$$

Using Equation (18):

$$= \log B(\boldsymbol{\alpha}) - \sum_{k=1}^{K} (\alpha_k - 1)\big(\psi(\alpha_k) - \psi(\alpha_0)\big)$$

$$= \log B(\boldsymbol{\alpha}) + \sum_{k=1}^{K} (\alpha_k - 1)\psi(\alpha_0) - \sum_{k=1}^{K} (\alpha_k - 1)\psi(\alpha_k)$$

$$= \log B(\boldsymbol{\alpha}) + (\alpha_0 - K)\psi(\alpha_0) - \sum_{k=1}^{K} (\alpha_k - 1)\psi(\alpha_k)$$

## B.3 Kullback-Leibler Divergence between two Dirichlets

The following result is presented using an adapted derivation by Lin (2016) and appears in Chen et al. (2018) and Joo et al. (2020) as a starting point for their variational objective (see Appendix C.7). In the following we use $\text{Dir}(\boldsymbol{\mu}; \boldsymbol{\alpha})$ to denote the optimized distribution, and $\text{Dir}(\boldsymbol{\mu}; \boldsymbol{\gamma})$ the reference or target distribution.

$$\text{KL}\left[p(\boldsymbol{\mu}|\boldsymbol{\alpha})\middle\|p(\boldsymbol{\mu}|\boldsymbol{\gamma})\right] = \mathbb{E}\left[\log \frac{p(\boldsymbol{\mu}|\boldsymbol{\alpha})}{p(\boldsymbol{\mu}|\boldsymbol{\gamma})}\right] = \mathbb{E}\left[\log p(\boldsymbol{\mu}|\boldsymbol{\alpha})\right] - \mathbb{E}\left[\log p(\boldsymbol{\mu}|\boldsymbol{\gamma})\right]$$

$$= \mathbb{E}\left[-\log B(\boldsymbol{\alpha}) + \sum_{k=1}^{K} (\alpha_k - 1) \log \mu_k\right]$$

$$-\mathbb{E}\left[-\log B(\boldsymbol{\gamma}) + \sum_{k=1}^{K} (\gamma_k - 1) \log \mu_k\right]$$

Distributing and pulling out $B(\boldsymbol{\alpha})$ and $B(\boldsymbol{\gamma})$ out of the expectation (they don't depend on $\boldsymbol{\mu}$):

$$= -\log \frac{B(\boldsymbol{\gamma})}{B(\boldsymbol{\alpha})} + \mathbb{E}\left[\sum_{k=1}^{K} (\alpha_k - 1) \log \mu_k - (\gamma_k - 1) \log \mu_k\right]$$

$$= -\log \frac{B(\boldsymbol{\gamma})}{B(\boldsymbol{\alpha})} + \mathbb{E}\left[\sum_{k=1}^{K}(\alpha_k - \gamma_k)\log\mu_k\right]$$

Moving the expectation inward and using the identity $\mathbb{E}[\mu_k] = \psi(\alpha_k) - \psi(\alpha_0)$ from Appendix B.1:

$$= -\log \frac{B(\boldsymbol{\gamma})}{B(\boldsymbol{\alpha})} + \sum_{k=1}^{K}(\alpha_k - \gamma_k)\big(\psi(\alpha_k) - \psi(\alpha_0)\big)$$

The KL divergence is also used by some works as regularizer by penalizing the distance to a uniform Dirichlet with $\boldsymbol{\gamma} = \mathbf{1}$ (Sensoy et al., 2018). In this case, the result above can be derived to be

$$\mathrm{KL}\left[p(\boldsymbol{\mu}|\boldsymbol{\alpha})\big|\big|p(\boldsymbol{\mu}|\mathbf{1})\right] = \log\frac{\Gamma(K)}{B(\boldsymbol{\alpha})} + \sum_{k=1}^{K}(\alpha_k - 1)\big(\psi(\alpha_k) - \psi(\alpha_0)\big)$$

where the $\log\Gamma(K)$ term can also be omitted for optimization purposes, since it does not depend on $\boldsymbol{\alpha}$.

## C   Additional Derivations Appendix

In this appendix we present relevant results in a Machine Learning context, including from some of the surveyed works, featuring as unified notation and annotated derivation steps. These include derivations of expected entropy (Appendix C.1) and mutual information (Appendix C.2) as uncertainty metrics for Dirichlet networks. Also, we derive a multitude of loss functions, including the $l_\infty$ norm loss of a Dirichlet w.r.t. a one-hot encoded class label in Appendix C.3, the $l_2$ norm loss in Appendix C.4, as well as the reverse KL loss by Malinin & Gales (2019), the UCE objective Biloš et al. (2019); Charpentier et al. (2020) and ELBO Shen et al. (2020); Chen et al. (2018) as training objectives (Appendices C.5 to C.7).

### C.1   Derivation of Expected Entropy

The following derivation is adapted from Malinin & Gales (2018) appendix section C.4. In the following, we assume that $\forall k \in \mathbb{K} : \mu_k > 0$:

$$\mathbb{E}_{p(\boldsymbol{\mu}|\mathbf{x},\hat{\boldsymbol{\theta}})}\Big[H\big[P(y|\boldsymbol{\mu})\big]\Big] = \int p(\boldsymbol{\mu}|\mathbf{x},\hat{\boldsymbol{\theta}})\Big(-\sum_{k=1}^{K}\mu_k\log\mu_k\Big)d\boldsymbol{\mu}$$

$$= -\sum_{k=1}^{K}\int p(\boldsymbol{\mu}|\mathbf{x},\hat{\boldsymbol{\theta}})\Big(\mu_k\log\mu_k\Big)d\boldsymbol{\mu}$$

Inserting the definition of $p(\boldsymbol{\mu}|\mathbf{x},\hat{\boldsymbol{\theta}}) \approx p(\boldsymbol{\mu}|\mathbf{x},\mathbb{D})$:

$$= -\sum_{k=1}^{K}\left(\frac{\Gamma(\alpha_0)}{\prod_{k'=1}^{K}\Gamma(\alpha_{k'})}\int\mu_k\log\mu_k\prod_{k'=1}^{K}\mu_{k'}^{\alpha_{k'}-1}d\boldsymbol{\mu}\right)$$

Singling out the factor $\mu_k$:

$$= -\sum_{k=1}^{K}\left(\frac{\Gamma(\alpha_0)}{\Gamma(\alpha_k)\prod_{k'\neq k}\Gamma(\alpha_{k'})}\mu_k^{\alpha_k-1}\int\mu_k\log\mu_k\prod_{k'\neq k}\mu_{k'}^{\alpha_{k'}-1}d\boldsymbol{\mu}\right)$$

Adjusting the normalizing constant (this is the same trick used in Appendix B.1):

$$= -\sum_{k=1}^{K}\left(\frac{\alpha_k}{\alpha_0}\int\frac{\Gamma(\alpha_0+1)}{\Gamma(\alpha_k+1)\prod_{k'\neq k}\Gamma(\alpha_{k'})}\mu_k^{\alpha_k-1}\log\mu_k\prod_{k'\neq k}\mu_{k'}^{\alpha_{k'}-1}d\boldsymbol{\mu}\right)$$

Using the identity $\mathbb{E}[\log\mu_k] = \psi(\alpha_k) - \psi(\alpha_0)$ (Equation (18)). Since the expectation here is w.r.t to a Dirichlet with concentration parameters $\alpha_k + 1$, we obtain

$$= -\sum_{k=1}^{K}\frac{\alpha_k}{\alpha_0}\Big(\psi(\alpha_k+1) - \psi(\alpha_0+1)\Big)$$

### C.2   Derivation of Mutual Information

We start from the expression in Equation (8):

$$I\Big[y,\boldsymbol{\mu}\,\Big|\,\mathbf{x},\mathbb{D}\Big] = H\bigg[\mathbb{E}_{p(\boldsymbol{\mu}|\mathbf{x},\mathbb{D})}\Big[P(y|\boldsymbol{\mu})\Big]\bigg] - \mathbb{E}_{p(\boldsymbol{\mu}|\mathbf{x},\mathbb{D})}\Big[H\big[P(y|\boldsymbol{\mu})\big]\Big]$$

Given that $\mathbb{E}[\mu_k] = \frac{\alpha_k}{\alpha_0}$ (Appendix B.1) and assuming that point estimate $p(\boldsymbol{\mu}\,|\,\mathbf{x}, \mathbb{D}) \approx p(\boldsymbol{\mu}\,|\,\mathbf{x}, \hat{\boldsymbol{\theta}})$ is sufficient (Malinin & Gales, 2018), we can identify the first term as the Shannon entropy $-\sum_{k=1}^{K} \mu_k \log \mu_k = -\sum_{k=1}^{K} \frac{\alpha_k}{\alpha_0} \log \frac{\alpha_k}{\alpha_0}$. Furthermore, the second part we already derived in Appendix C.1 and thus we obtain:

$$= -\sum_{k=1}^{K} \frac{\alpha_k}{\alpha_0} \log \frac{\alpha_k}{\alpha_0} + \sum_{k=1}^{K} \frac{\alpha_k}{\alpha_0} \left( \psi(\alpha_k + 1) - \psi(\alpha_0 + 1) \right)$$

$$= -\sum_{k=1}^{K} \frac{\alpha_k}{\alpha_0} \left( \log \frac{\alpha_k}{\alpha_0} - \psi(\alpha_k + 1) + \psi(\alpha_0 + 1) \right)$$

### C.3 $l_\infty$ Norm Derivation

In this section we elaborate on the derivation of Tsiligkaridis (2019) deriving a generalized $l_p$ loss, upper-bounding the $l_\infty$ loss. This in turn allows us to easily derive the $l_2$ loss used by Sensoy et al. (2018); Zhao et al. (2020). Here we assume the classification target $y$ is provided in the form of a one-hot encoded label $\mathbf{y} = [\mathbf{1}_{y=1}, \ldots, \mathbf{1}_{y=K}]^T$.

$$\mathbb{E}_{p(\boldsymbol{\mu}\,|\,\mathbf{x}, \boldsymbol{\theta})}\left[|| \mathbf{y} - \boldsymbol{\mu} ||_\infty\right] \leq \mathbb{E}_{p(\boldsymbol{\mu}\,|\,\mathbf{x}, \boldsymbol{\theta})}\left[|| \mathbf{y} - \boldsymbol{\mu} ||_p\right] \tag{19}$$

Using Jensen's inequality

$$\leq \left( \mathbb{E}_{p(\boldsymbol{\mu}\,|\,\mathbf{x}, \boldsymbol{\theta})}\left[|| \mathbf{y} - \boldsymbol{\mu} ||_p^p\right] \right)^{1/p} \tag{20}$$

Evaluating the expression with $\forall k \neq y : \mathbf{y}_k = 0$:

$$= \left( \mathbb{E}[(1 - \mu_y)^p] + \sum_{k \neq y} \mathbb{E}[\mu_k^p] \right)^{1/p} \tag{21}$$

In order to compute the expression above, we first realize that all components of $\mu$ are distributed according to a Beta distribution $\text{Beta}(\alpha, \beta)$ (since the Dirichlet is a multivariate generalization of the beta distribution) for which the moment-generating function is given as follows:

$$\mathbb{E}[\mu^p] = \frac{\Gamma(\alpha + p)\Gamma(\beta)\Gamma(\alpha + \beta)}{\Gamma(\alpha + p + \beta)\Gamma(\alpha)\Gamma(\beta)} = \frac{\Gamma(\alpha + p)\Gamma(\alpha + \beta)}{\Gamma(\alpha + p + \beta)\Gamma(\alpha)}$$

Given that the first term in Equation (19) is characterized by $\text{Beta}(\alpha_0 - \alpha_y, \alpha_y)$ and the second one by $\text{Beta}(\alpha_k, \alpha_0 - \alpha_k)$, we can evaluate the result in Equation (19) using the moment generating function:

$$\mathbb{E}_{p(\boldsymbol{\mu}\,|\,\mathbf{x}, \boldsymbol{\theta})}\left[|| \mathbf{y} - \boldsymbol{\mu} ||_\infty\right] \leq \left( \frac{\Gamma(\alpha_0 - \alpha_y + p)\Gamma(\alpha_0 - \cancel{\alpha_y} + \cancel{\alpha_y})}{\Gamma(\alpha_0 - \cancel{\alpha_y} + p + \cancel{\alpha_y})\Gamma(\alpha_0 - \alpha_y)} + \sum_{k \neq y} \frac{\Gamma(\alpha_k + p)\Gamma(\cancel{\alpha_k} + \alpha_0 - \cancel{\alpha_k})}{\Gamma(\cancel{\alpha_k} + p + \alpha_0 - \cancel{\alpha_k})\Gamma(\alpha_k)} \right)^{\frac{1}{p}}$$

$$= \left( \frac{\Gamma(\alpha_0 - \alpha_y + p)\Gamma(\alpha_0)}{\Gamma(\alpha_0 + p)\Gamma(\alpha_0 - \alpha_y)} + \sum_{k \neq y} \frac{\Gamma(\alpha_k + p)\Gamma(\alpha_0)}{\Gamma(p + \alpha_0)\Gamma(\alpha_k)} \right)^{\frac{1}{p}}$$

Factoring out common terms:

$$= \left( \frac{\Gamma(\alpha_0)}{\Gamma(\alpha_0 + p)} \left( \frac{\Gamma(\alpha_0 - \alpha_y + p)}{\Gamma(\alpha_0 - \alpha_y)} + \sum_{k \neq y} \frac{\Gamma(\alpha_k + p)}{\Gamma(\alpha_k)} \right) \right)^{\frac{1}{p}}$$

Expressing $\alpha_0 - \alpha_k = \sum_{k \neq y} \alpha_k$:

$$= \left( \frac{\Gamma(\alpha_0)}{\Gamma(\alpha_0 + p)} \right)^{\frac{1}{p}} \left( \frac{\Gamma\left( \sum_{k \neq y} \alpha_k + p \right)}{\Gamma\left( \sum_{k \neq y} \alpha_k \right)} + \sum_{k \neq y} \frac{\Gamma(\alpha_k + p)}{\Gamma(\alpha_k)} \right)^{\frac{1}{p}}$$

### C.4 $l_2$ Norm Loss Derivation

Here we present an adapted derivation by Sensoy et al. (2018) for the $l_2$-norm loss to train Dirichlet networks. Here we again use a one-hot vector for a label with $\mathbf{y} = [\mathbf{1}_{y=1}, \ldots, \mathbf{1}_{y=K}]^T$.

$$\mathbb{E}_{p(\boldsymbol{\mu} \mid \mathbf{x}, \boldsymbol{\theta})} \left[ \| \mathbf{y} - \boldsymbol{\mu} \|_2^2 \right] = \mathbb{E} \left[ \sum_{k=1}^{K} (\mathbf{1}_{y=k} - \mu_k)^2 \right] \tag{22}$$

$$= \mathbb{E} \left[ \sum_{k=1}^{K} \mathbf{1}_{y=k}^2 - 2\mu_k \mathbf{1}_{y=k} + \mu_k^2 \right] \tag{23}$$

$$= \sum_{k=1}^{K} \mathbf{1}_{y=k}^2 - 2\mathbb{E}[\mu_k]\mathbf{1}_{y=k} + \mathbb{E}[\mu_k^2] \tag{24}$$

Using the identity that $\mathbb{E}[\mu_k^2] = \mathbb{E}[\mu_k]^2 + \text{Var}(\mu_k)$:

$$= \sum_{k=1}^{K} \mathbf{1}_{y=k}^2 - 2\mathbb{E}[\mu_k]\mathbf{1}_{y=k} + \mathbb{E}[\mu_k]^2 + \text{Var}(\mu_k) \tag{25}$$

$$= \sum_{k=1}^{K} \left( \mathbf{1}_{y=k} - \mathbb{E}[\mu_k] \right)^2 + \text{Var}(\mu_k) \tag{26}$$

Finally, we use the result from Appendix B.1 and the result that $\text{Var}(\mu_k) = \dfrac{\alpha_k(\alpha_0 - \alpha_k)}{\alpha_0^2(\alpha_0 + 1)}$ (see Lin, 2016):

$$= \sum_{k=1}^{K} \left( \mathbf{1}_{y=k} - \frac{\alpha_k}{\alpha_0} \right)^2 + \frac{\alpha_k(\alpha_0 - \alpha_k)}{\alpha_0^2(\alpha_0 + 1)} \tag{27}$$

### C.5 Derivation of Reverse KL loss

Here we re-state and annotate the derivation of reverse KL loss by Malinin & Gales (2019) in more detail, starting from the forward KL loss by Malinin & Gales (2018). Note that here, $\hat{\boldsymbol{\alpha}}$ contains a dependence on $k$, since Malinin & Gales (2018) let $\hat{\alpha}_k = \hat{\mu}_k \hat{\alpha}_0$ with $\hat{\alpha}_0$ being a hyperparameter and $\hat{\mu}_k = \mathbf{1}_{k=y} + (-\mathbf{1}_{k=y}K + 1)\varepsilon$ and $\varepsilon$ being a small number.

$$\mathbb{E}_{p(\mathbf{x},y)} \left[ \sum_{k=1}^{K} \mathbf{1}_{y=k} \text{KL} \left[ p(\boldsymbol{\mu}|\hat{\boldsymbol{\alpha}}) \middle\| p(\boldsymbol{\mu}| \mathbf{x}, \boldsymbol{\theta}) \right] \right]$$

$$= \mathbb{E}_{p(\mathbf{x},y)} \left[ \sum_{k=1}^{K} \mathbf{1}_{y=k} \int p(\boldsymbol{\mu}|\hat{\boldsymbol{\alpha}}) \log \frac{p(\boldsymbol{\mu}|\hat{\boldsymbol{\alpha}})}{p(\boldsymbol{\mu}| \mathbf{x}, \boldsymbol{\theta})} d\boldsymbol{\mu} \right]$$

Writing the expectation explicitly:

$$= \int \sum_{k=1}^{K} p(y = k, \mathbf{x}) \sum_{k=1}^{K} \mathbf{1}_{y=k} \int p(\boldsymbol{\mu}|\hat{\boldsymbol{\alpha}}) \log \frac{p(\boldsymbol{\mu}|\hat{\boldsymbol{\alpha}})}{p(\boldsymbol{\mu}| \mathbf{x}, \boldsymbol{\theta})} d\boldsymbol{\mu} d\mathbf{x}$$

$$= \int \sum_{k=1}^{K} p(\mathbf{x}) P(y = k | \mathbf{x}) \sum_{k=1}^{K} \mathbf{1}_{y=k} \int p(\boldsymbol{\mu}|\hat{\boldsymbol{\alpha}}) \log \frac{p(\boldsymbol{\mu}|\hat{\boldsymbol{\alpha}})}{p(\boldsymbol{\mu}|\mathbf{x}, \boldsymbol{\theta})} d\boldsymbol{\mu} d\mathbf{x}$$

$$= \mathbb{E}_{p(\mathbf{x})} \left[ \sum_{k=1}^{K} P(y = k | \mathbf{x}) \sum_{k=1}^{K} \mathbf{1}_{y=k} \int p(\boldsymbol{\mu}|\hat{\boldsymbol{\alpha}}) \log \frac{p(\boldsymbol{\mu}|\hat{\boldsymbol{\alpha}})}{p(\boldsymbol{\mu}|\mathbf{x}, \boldsymbol{\theta})} d\boldsymbol{\mu} \right]$$

Adding factor in log, collapsing double sum:

$$= \mathbb{E}_{p(\mathbf{x})} \left[ \sum_{k=1}^{K} P(y = k | \mathbf{x}) \int p(\boldsymbol{\mu}|\hat{\boldsymbol{\alpha}}) \log \left( \frac{p(\boldsymbol{\mu}|\hat{\boldsymbol{\alpha}}) \sum_{k=1}^{K} P(y = k | \mathbf{x})}{p(\boldsymbol{\mu}|\mathbf{x}, \boldsymbol{\theta}) \sum_{k=1}^{K} P(y = k | \mathbf{x})} \right) d\boldsymbol{\mu} \right]$$

Reordering, separating constant factor from log:

$$= \mathbb{E}_{p(\mathbf{x})} \left[ \int \sum_{k=1}^{K} P(y = k | \mathbf{x}) p(\boldsymbol{\mu}|\hat{\boldsymbol{\alpha}}) \left( \log \left( \frac{\sum_{k=1}^{K} P(y = k | \mathbf{x}) p(\boldsymbol{\mu}|\hat{\boldsymbol{\alpha}})}{p(\boldsymbol{\mu}|\mathbf{x}, \boldsymbol{\theta})} \right) \right. \right.$$
$$\left. \left. - \underbrace{\log \left( \sum_{k=1}^{K} P(y = k | \mathbf{x}) \right)}_{=0} \right) d\boldsymbol{\mu} \right]$$

$$= \mathbb{E}_{p(\mathbf{x})} \left[ \mathrm{KL} \left[ \underbrace{\sum_{k=1}^{K} P(y = k | \mathbf{x}) p(\boldsymbol{\mu}|\hat{\alpha})}_{\text{Mixture of } K \text{ Dirichlets}} \middle\| p(\boldsymbol{\mu}|\mathbf{x}, \boldsymbol{\theta}) \right] \right]$$

where we can see that this objective actually tries to minimizes the divergence towards a mixture of $K$ Dirichlet distributions. In the case of high data uncertainty, this is claimed incentivize the model to distribute mass around each of the corners of the simplex, instead of the desired behavior shown in Figure 2c. Therefore, Malinin & Gales (2019) propose to swap the order of arguments in the KL-divergence, resulting in the following:

$$\mathbb{E}_{p(\mathbf{x})} \left[ \sum_{k=1}^{K} P(y = k | \mathbf{x}) \cdot \mathrm{KL} \left[ p(\boldsymbol{\mu}|\mathbf{x}, \boldsymbol{\theta}) \middle\| p(\boldsymbol{\mu}|\hat{\boldsymbol{\alpha}}) \right] \right]$$

$$= \mathbb{E}_{p(\mathbf{x})} \left[ \sum_{k=1}^{K} p(y = k | \mathbf{x}) \cdot \int p(\boldsymbol{\mu}|\mathbf{x}, \boldsymbol{\theta}) \log \frac{p(\boldsymbol{\mu}|\mathbf{x}, \boldsymbol{\theta})}{p(\boldsymbol{\mu}|\hat{\boldsymbol{\alpha}})} d\boldsymbol{\mu} \right]$$

Reordering:

$$= \mathbb{E}_{p(\mathbf{x})} \left[ \int p(\boldsymbol{\mu}|\mathbf{x}, \boldsymbol{\theta}) \sum_{k=1}^{K} P(y = k | \mathbf{x}) \log \frac{p(\boldsymbol{\mu}|\mathbf{x}, \boldsymbol{\theta})}{p(\boldsymbol{\mu}|\hat{\boldsymbol{\alpha}})} d\boldsymbol{\mu} \right]$$

$$= \mathbb{E}_{p(\mathbf{x})} \left[ \mathbb{E}_{p(\boldsymbol{\mu}|\mathbf{x}, \boldsymbol{\theta})} \left[ \sum_{k=1}^{K} P(y = k | \mathbf{x}) \log p(\boldsymbol{\mu}|\mathbf{x}, \boldsymbol{\theta}) - \sum_{k=1}^{K} P(y = k | \mathbf{x}) \log p(\boldsymbol{\mu}|\hat{\boldsymbol{\alpha}}) \right] \right]$$

$$= \mathbb{E}_{p(\mathbf{x})} \left[ \int p(\boldsymbol{\mu}|\mathbf{x}, \boldsymbol{\theta}) \left( \log \left( \prod_{k=1}^{K} p(\boldsymbol{\mu}|\mathbf{x}, \boldsymbol{\theta})^{P(y=k|\mathbf{x})} \right) - \log \left( \prod_{k=1}^{K} p(\boldsymbol{\mu}|\hat{\boldsymbol{\alpha}})^{P(y=k|\mathbf{x})} \right) \right) d\boldsymbol{\mu} \right]$$

$$= \mathbb{E}_{p(\mathbf{x})} \left[ \int p(\boldsymbol{\mu}|\mathbf{x}, \boldsymbol{\theta}) \left( \log \left( p(\boldsymbol{\mu}|\mathbf{x}, \boldsymbol{\theta})^{\sum_{k=1}^{K} P(y=k|\mathbf{x})} \right) \right. \right.$$
$$\left. \left. - \log \left( \prod_{k=1}^{K} \left( \frac{1}{B(\boldsymbol{\alpha})} \prod_{k'=1}^{K} \mu_{k'}^{\alpha_{k'}-1} \right)^{p(y=k|\mathbf{x})} \right) \right) d\boldsymbol{\mu} \right]$$

$$
= \mathbb{E}_{p(\mathbf{x})}\left[\int p(\boldsymbol{\mu}|\mathbf{x},\boldsymbol{\theta})\left(\log\left(p(\boldsymbol{\mu}|\mathbf{x},\boldsymbol{\theta})\right) - \log\left(\prod_{k=1}^{K}\left(\frac{1}{B(\boldsymbol{\alpha})}\prod_{k'=1}^{K}\mu_{k'}^{\alpha_{k'}-1}\right)^{P(y=k|\mathbf{x})}\right)d\boldsymbol{\mu}\right)\right]
$$

$$
= \mathbb{E}_{p(\mathbf{x})}\left[\int p(\boldsymbol{\mu}|\mathbf{x},\boldsymbol{\theta})\left(\log\left(p(\boldsymbol{\mu}|\mathbf{x},\boldsymbol{\theta})\right) - \log\left(\frac{1}{B(\boldsymbol{\alpha})}\prod_{k'=1}^{K}\mu_{k'}^{\sum_{k=1}^{K}P(y=k|\mathbf{x})\alpha_{k'}-1}\right)d\boldsymbol{\mu}\right)\right]
$$

$$
= \mathbb{E}_{p(\mathbf{x})}\left[\mathrm{KL}\Big[p(\boldsymbol{\mu}|\mathbf{x},\boldsymbol{\theta})||p(\boldsymbol{\mu}|\bar{\boldsymbol{\alpha}})\Big]\right] \quad \text{where} \quad \bar{\boldsymbol{\alpha}} = \sum_{k=1}^{K}p(y=k|\mathbf{x})\alpha_{k'}
$$

Therefore, instead of a mixture of Dirichlet distribution, we obtain a single distribution whose *parameters are a mixture* of the concentrations of each class.

### C.6   Uncertainty-aware Cross-Entropy Loss

The uncertainty-aware cross-entropy loss in Biloš et al. (2019); Charpentier et al. (2020) has the form

$$
\mathcal{L}_{\mathrm{UCE}} = \mathbb{E}_{p(\boldsymbol{\mu}|\mathbf{x},\boldsymbol{\theta})}[\log p(y|\boldsymbol{\mu})] = \mathbb{E}[\log\mu_y] = \psi(\alpha_y) - \psi(\alpha_0)
$$

as $p(y|\boldsymbol{\mu})$ is given by the true label in form of a delta distribution, we can apply the result from Appendix B.1.

### C.7   Evidence-Lower Bound For Dirichlet Posterior Estimation

The evidence lower bound is a well-known objective to optimize the KL-divergence between an approximate proposal and target distribution (Jordan et al., 1999; Kingma & Welling, 2014). We derive it based on Chen et al. (2018) in the following for the Dirichlet case with a proposal distribution $p(\boldsymbol{\mu}|\mathbf{x},\boldsymbol{\theta})$ to the target distribution $p(\boldsymbol{\mu}|y)$. For the first part of the derivation, we omit the dependence on $\boldsymbol{\beta}$ for clarity.

$$
\mathrm{KL}\big[p(\boldsymbol{\mu}|\mathbf{x},\boldsymbol{\theta})\big|\big|p(\boldsymbol{\mu}|y)\big] = \mathbb{E}_{p(\boldsymbol{\mu}|\mathbf{x},\boldsymbol{\theta})}\left[\log\frac{p(\boldsymbol{\mu}|\mathbf{x},\boldsymbol{\theta})}{p(\boldsymbol{\mu}|y)}\right] = \mathbb{E}_{p(\boldsymbol{\mu}|\mathbf{x},\boldsymbol{\theta})}\left[\log\frac{p(\boldsymbol{\mu}|\mathbf{x},\boldsymbol{\theta})p(y)}{p(\boldsymbol{\mu},y)}\right]
$$

Factorizing $p(\boldsymbol{\mu},y) = P(y|\boldsymbol{\mu})p(\boldsymbol{\mu})$, pulling out $p(y)$ as it doesn't depend on $\mu$:

$$
= \mathbb{E}_{p(\boldsymbol{\mu}|\mathbf{x},\boldsymbol{\theta})}\left[\log\frac{p(\boldsymbol{\mu}|\mathbf{x},\boldsymbol{\theta})}{P(y|\boldsymbol{\mu})p(\boldsymbol{\mu})}\right] + p(y)
$$

$$
= \mathbb{E}_{p(\boldsymbol{\mu}|\mathbf{x},\boldsymbol{\theta})}\left[\log\frac{p(\boldsymbol{\mu}|\mathbf{x},\boldsymbol{\theta})}{p(\boldsymbol{\mu})}\right] - \mathbb{E}_{p(\boldsymbol{\mu}|\mathbf{x},\boldsymbol{\theta})}\big[\log P(y|\boldsymbol{\mu})\big] + p(y)
$$

$$
\leq \mathrm{KL}\big[p(\boldsymbol{\mu}|\mathbf{x},\boldsymbol{\theta})\big|\big|p(\boldsymbol{\mu})\big] - \mathbb{E}_{p(\boldsymbol{\mu}|\mathbf{x},\boldsymbol{\theta})}\big[\log P(y|\boldsymbol{\mu})\big]
$$

Now note that the second part of the result is the uncertainty-aware cross-entropy loss from Appendix C.6 and re-adding the dependence of $p(\mu)$ on $\boldsymbol{\gamma}$, we can re-use our result regarding the KL-divergence between two Dirichlets in Appendix B.3 and thus obtain:

$$
\mathcal{L}_{\mathrm{ELBO}} = \psi(\beta_y) - \psi(\beta_0) - \log\frac{B(\boldsymbol{\beta})}{B(\boldsymbol{\gamma})} + \sum_{k=1}^{K}(\beta_k - \gamma_k)\big(\psi(\beta_k) - \psi(\beta_0)\big) \tag{28}
$$

which is exactly the solution obtained by both Chen et al. (2018) and Joo et al. (2020).

## D  Overview over Loss Functions Appendix

In Tables 5 and 6, we compare the forms of the loss function used by Evidential Deep Learning methods for classification, using the consistent notation from the paper. Most of the presented results can be found in the previous Appendix B and Appendix C. We refer to the original work for details about the objective of Nandy et al. (2020).

Table 5: Overview over objectives used by prior networks for classification.

| Method | Loss function | Regularizer | Comment |
|---|---|---|---|
| Prior networks (Malinin & Gales, 2018) | $\log \frac{B(\hat{\boldsymbol{\alpha}})}{B(\boldsymbol{\alpha})} + \sum_{k=1}^{K}(\alpha_k - \hat{\alpha}_k)\big(\psi(\alpha_k) - \psi(\alpha_0)\big)$ | $-\log \frac{\Gamma(K)}{B(\boldsymbol{\alpha})} + \sum_{k=1}^{K}(\alpha_k - 1)(\psi(\alpha_k) - \psi(\alpha_0))$ | Target concentration parameters $\hat{\boldsymbol{\alpha}}$ are created using a label smoothing approach, i.e. $\hat{\mu}_k = \begin{cases} 1 - (K-1)\varepsilon & \text{if } y = k \\ \varepsilon & \text{if } y \neq k \end{cases}$. Together with setting $\hat{\alpha}_0$ as a hyperparameter, $\hat{\alpha}_k = \hat{\mu}_k \hat{\alpha}_0$ |
| Prior networks (Malinin & Gales, 2019) | $\log \frac{B(\hat{\boldsymbol{\alpha}})}{B(\boldsymbol{\alpha})} + \sum_{k=1}^{K}(\alpha_k - \hat{\alpha}_k)\big(\psi(\alpha_k) - \psi(\alpha_0)\big)$ | $\log \frac{B(\bar{\boldsymbol{\alpha}})}{B(\boldsymbol{\alpha})} + \sum_{k=1}^{K}(\alpha_k - \bar{\alpha}_k)\big(\psi(\alpha_k) - \psi(\alpha_0)\big)$ | Similar to above, $\hat{\alpha}_c^{(k)} = \mathbf{1}_{c=k}\alpha_{\text{in}} + 1$ for in-distribution and $\bar{\alpha}_c^{(k)} = \mathbf{1}_{c=k}\alpha_{\text{out}} + 1$ where we have hyperparameters set to $\alpha_{\text{in}} = 0.01$ and $\alpha_{\text{out}} = 0$. Then finally, $\hat{\boldsymbol{\alpha}} = \sum_{k=1}^{K} p(y = k \mid \mathbf{x}) \hat{\alpha}_k$ and $\bar{\boldsymbol{\alpha}} = \sum_{k=1}^{K} p(y = k \mid \mathbf{x}) \bar{\alpha}_k$. |
| Information Robust Dirichlet Networks (Tsiligkaridis, 2019) | $\left(\frac{\Gamma(\alpha_0)}{\Gamma(\alpha_0 + p)}\right)^{\frac{1}{p}} \left( \frac{\Gamma\left(\sum_{k \neq y} \alpha_k + p\right)}{\Gamma\left(\sum_{k \neq y} \alpha_k\right)} + \sum_{k \neq y} \frac{\Gamma(\alpha_k + p)}{\Gamma(\alpha_k)} \right)^{\frac{1}{p}}$ | $\frac{1}{2}\sum_{k \neq y}(\alpha_k - 1)^2 (\psi^{(1)}(\alpha_k) - \psi^{(1)})(\alpha_0))$ | $\psi^{(1)}$ is the polygamma function defined as $\psi^{(1)}(x) = \frac{d}{dx}\psi(x)$. |
| Dirichlet via Function Decomposition (Biloš et al., 2019) | $\psi(\alpha_y) - \psi(\alpha_0)$ | $\lambda_1 \int_0^T \mu_k(\tau)^2 d\tau + \lambda_2 \int_0^T (\nu - \sigma^2(\tau))^2 d\tau$ | Factors $\lambda_1$ and $\lambda_2$ that are treated as hyperparameters that weigh first term pushing the for logit $k$ to zero, while pushing the variance in the first term to $\nu$. |
| Prior network with PAC Reg. (Haussmann et al., 2019) | $-\log \mathbb{E}\left[\prod_{k=1}^{K}\left(\frac{\alpha_k}{\alpha_0}\right)^{\mathbf{1}_{k=y}}\right]$ | $\sqrt{\frac{\text{KL}\left[p(\boldsymbol{\mu}\mid\boldsymbol{\alpha}) \,\middle\|\, p(\boldsymbol{\mu}\mid\mathbf{1})\right] - \log \delta}{N} - 1}$ | The expectation in the loss function is evaluated using parameter samples from a weight distribution. $\delta \in [0, 1]$. |
| Ensemble Distribution Distillation (Malinin et al., 2020b) | $\psi(\alpha_0) - \sum_{k=1}^{K}\psi(\alpha_k) + \frac{1}{M}\sum_{m=1}^{M}\sum_{k=1}^{K}(\alpha_k - 1)$ $\log p(y = k \mid \mathbf{x}, \boldsymbol{\theta}^{(m)})$ | - | The objective uses predictions from a trained ensemble with parameters $\boldsymbol{\theta}_1, \ldots, \boldsymbol{\theta}_M$. |
| Prior networks with representation gap (Nandy et al., 2020) | $-\log \mu_y - \frac{\lambda_{\text{in}}}{K}\sum_{k=1}^{K}\sigma(\alpha_k)$ | $-\sum_{k=1}\frac{1}{K}\log \mu_k - \frac{\lambda_{\text{out}}}{K}\sum_{k=1}^{K}\sigma(\alpha_k)$ | The main objective is being optimized on in-distribution, the regularizer on out-of-distribution data. $\lambda_{\text{in}}$ and $\lambda_{\text{out}}$ weighing terms and $\sigma$ denotes the sigmoid function. |
| Prior RNN (Shen et al., 2020) | $\sum_{k=1}\mathbf{1}_{k=y}\log \mu_k$ | $-\log B(\tilde{\boldsymbol{\alpha}}) + (\hat{\alpha}_0 - K)\psi(\hat{\alpha}_0) - \sum_{k=1}^{K}(\hat{\alpha}_k - 1)\psi(\hat{\alpha}_k)$ | Here, the entropy regularizer operates on a scaled version of the concentration parameters $\tilde{\boldsymbol{\alpha}} = (\mathbf{I}_K - \mathbf{W})\boldsymbol{\alpha}$, where $\mathbf{W}$ is learned. |
| Graph-based Kernel Dirichlet dist. est. (GKDE) (Zhao et al., 2020) | $\sum_{k=1}^{K}\left(\mathbf{1}_{y=k} - \frac{\alpha_k}{\alpha_0}\right)^2 + \frac{\alpha_k(\alpha_0 - \alpha_k)}{\alpha_0^2(\alpha_0 + 1)}$ | $-\log \frac{B(\boldsymbol{\alpha})}{B(\hat{\boldsymbol{\alpha}})} + \sum_{k=1}^{K}(\alpha_k - \hat{\alpha}_k)\big(\psi(\alpha_k) - \psi(\alpha_0)\big)$ | $\hat{\boldsymbol{\alpha}}$ here corresponds to a uniform prior including some information about the local graph structure. The authors also use an additional knowledge distillation objective, which was omitted here since it doesn't related to the Dirichlet. |

Table 6: Overview over objectives used by posterior networks for classification.

| Method | Loss function | Regularizer | Comment |
|---|---|---|---|
| Evidential Deep Learning (Sensoy et al., 2018) | $\sum_{k=1}^{K}\left(\mathbf{1}_{y=k}-\frac{\beta_k}{\beta_0}\right)^2+\frac{\beta_k(\beta_0-\beta_k)}{\beta_0^2(\beta_0+1)}$ | $-\log\frac{\Gamma(K)}{B(\boldsymbol{\beta})}+\sum_{k=1}^{K}(\beta_k-1)(\psi(\beta_k)-\psi(\beta_0))$ | |
| Variational Dirichlet (Chen et al., 2018) | $\psi(\beta_y)-\psi(\beta_0)$ | $-\log\frac{B(\boldsymbol{\beta})}{B(\boldsymbol{\gamma})}+\sum_{k=1}^{K}(\beta_k-\gamma_k)\big(\psi(\beta_k)-\psi(\beta_0)\big)$ | |
| Regularized ENN Zhao et al. (2019) | $\sum_{k=1}^{K}\left(\mathbf{1}_{y=k}-\frac{\beta_k}{\beta_0}\right)^2+\frac{\beta_k(\beta_0-\beta_k)}{\beta_0^2(\beta_0+1)}$ | $-\lambda_1\mathbb{E}_{p_{\text{out}}(\mathbf{x},y)}\left[\frac{\alpha_y}{\alpha_0}\right]-\lambda_2\mathbb{E}_{p_{\text{confl.}}(\mathbf{x},y)}\left[\sum_{k=1}^{K}\left(\frac{\beta_k\sum_{k'\neq k}\beta_{k'}\left(1-\frac{|\beta_{k'}-\beta_k|}{\beta_{k'}+\beta_k}\right)}{\sum_{k'\neq k}\beta_{k'}}\right)\right]$ | The first term represents *vacuity*, i.e. the lack of evidence and is optimized using OOD examples. The second term stands for *dissonance*, and is computed using points with neighborhoods with different classes from their own. $\lambda_1,\lambda_2$ are hyperparameters. |
| WGAN–ENN (Hu et al., 2021) | $\sum_{k=1}^{K}\left(\mathbf{1}_{y=k}-\frac{\beta_k}{\beta_0}\right)^2+\frac{\beta_k(\beta_0-\beta_k)}{\beta_0^2(\beta_0+1)}$ | $-\lambda\mathbb{E}_{p_{\text{out}}(\mathbf{x},y)}\left[\frac{\alpha_y}{\alpha_0}\right]$ | |
| Belief Matching (Joo et al., 2020) | $\psi(\beta_y)-\psi(\beta_0)$ | $-\log\frac{B(\boldsymbol{\beta})}{B(\boldsymbol{\gamma})}+\sum_{k=1}^{K}(\beta_k-\gamma_k)\big(\psi(\beta_k)-\psi(\beta_0)\big)$ | |
| Posterior networks (Charpentier et al., 2020) | $\psi(\beta_y)-\psi(\beta_0)$ | $-\log B(\boldsymbol{\beta})+(\beta_0-K)\psi(\beta_0)-\sum_{k=1}^{K}(\beta_k-1)\psi(\beta_k)$ | |
| Graph Posterior Networks (Stadler et al., 2021) | $\psi(\beta_y)-\psi(\beta_0)$ | $-\log B(\boldsymbol{\beta})+(\beta_0-K)\psi(\beta_0)-\sum_{k=1}^{K}(\beta_k-1)\psi(\beta_k)$ | |
| Generative Evidential Neural Network (Sensoy et al., 2020) | $-\sum_{k=1}^{K}\left(\mathbb{E}_{p_{\text{in}}(\mathbf{x})}\big[\log(\sigma(f_{\boldsymbol{\theta}}(\mathbf{x})))\big]+\mathbb{E}_{p_{\text{out}}(\mathbf{x})}\big[\log(1-\sigma(f_{\boldsymbol{\theta}}(\mathbf{x})))\big]\right)$ | $-\log\frac{\Gamma(K)}{B(\boldsymbol{\beta}_{-y})}+\sum_{k\neq y}(\beta_k-1)(\psi(\beta_k)-\psi(\beta_0))$ | The main loss is a discriminative loss using ID and OOD samples, generated by a VAE. The regularizer is taken over all classes *excluding* the true class $y$ (also indicated by $\boldsymbol{\beta}_{-y}$). |

