# OpenReview forum: "Prior and Posterior Networks: A Survey on Evidential Deep Learning Methods For Uncertainty Estimation"
_TMLR — Rejected by TMLR_

### Review · Reviewer_jdKi · 2022-07-22

**Summary Of Contributions:**

This paper reviews recent advances in single-model, single-pass networks for predictive uncertainty quantification.
Specifically, the models considered are networks with Dirichlet output---such as prior networks, posterior networks, and evidential deep learning (EDL)---which have a higher degree of freedom in terms of uncertainty quantification due to the fact that the Dirichlet can be seen as a distribution over softmax outputs of standard NNs.
Additionally, models for regression are also discussed.

**Requested Changes:**

## Critical

1. I urge the authors to not constrain themselves with just 12 pages of main contents. The authors should take advantage of the fact that TMLR is a journal. An example of excellent review paper is [1]: it has ~50 pages of main contents and it's considered a one-stop reference for normalizing flows. I think it's crucial for the authors to follow their example, not only for the success of this paper but also for the success of the EDL class of models in general.
    1. One thing that the authors should do is to expand the discussion in Sec. 3.3. I think it should be longer and more extensive, with more math equations for clarity.
    2. Another thing to add is an experiment section, comparing different EDL models between themselves and against traditional UQ methods such as BNNs (e.g. [2]) and ensemble methods.
2. The discussion about different types of uncertainty is unconvincing to me.
    1. Is the epistemic uncertainty obtained via PNs really an epistemic uncertainty, e.g. is it reducible given more data? More extensive discussion would be great.
    2. Eq. 8 looks similar to the standard definition of epistemic uncertainty in BNNs (see e.g. [4]), but here it is called "distributional" uncertainty. More explanation is needed here.



## Suggestions

1. The paper will be **much** stronger if the authors, in addition to this paper, also propose an all-in-one software library for practitioners to easily use EDL for their problems. See, for example, [2].


## Minor

0. Last sentence of page 1: "... they fall back to the prior ..." is this the defining property of EDL? It doesn't seem so to me because other, non-EDL methods have this property, e.g. GPs, RBF networks.
1. Fig. 2 is hard to understand, e.g. why is (d) model uncertainty and (c) data. Please provide more explanation and add a colorbar.
2. Notational inconsistencies, e.g. both $P$ and $p$ are used to represent distributions without further explanation.
3. Footnote 3 is crucial since it differentiates PNs from standard NNs. It should be discussed in the main text.
4. Sec. 3.2. $\bf{\beta}$ is undefined and doesn't seem to be used anywhere.
5. Page 7: "... instead explicitly maximize the KL ..."---shouldn't it be "minimize"?
6. Table 1: what is "the approach of Liang et al"? Need an explicit explanation.
7. The writing needs to be more consistent, e.g. two kind of em-dashes are used `--` and `---`.
8. Restate the definition of $p(\mu \mid x, \theta)$ and $p(y \mid \mu)$ before eq. 7.
9. The "References" section is unpolished. Additional work on top of copy-pasting from Google Scholar is always needed, unfortunately. I suggest the authors polish their bibtex file: e.g. use proper capitalization (e.g. bayesian -> {B}ayesian), make the format uniform (e.g. should page number, location, etc. be included?), and check the latest venue of each paper (i.e. don't use the ArXiv version if it's published).
10. Separate the appendices from the main text. Add `\clearpage` in between and use "Appendix" in section titles, e.g. "Appendix A  Some Section" instead of "A Some Section".


### References
[1] Papamakarios, George, et al. "Normalizing flows for probabilistic modeling and inference." JMLR, 2021.

[2] https://github.com/aleximmer/laplace

[3] Daxberger, Kristiadi, Immer, Eschenhagen, Bauer, Hennig. "Laplace redux---effortless Bayesian deep learning." NeurIPS, 2021.

[4] Depeweg, Stefan, et al. "Decomposition of uncertainty in Bayesian deep learning for efficient and risk-sensitive learning." ICML, 2018.


**Strengths And Weaknesses:**

## Strengths

1. The paper is generally well written.
2. Almost all published works in this space are included.

## Weaknesses

1. The content and discussion are very sparse for a journal review paper. Specifically, Sec. 3.3 reads more like a "Related Work" section since the discussion for each method is very brief. Because of this, unfortunately, I can't say that I'm "up to speed" on this topic after reading the paper---which is my prior expectation for a review paper.
2. One of the main objectives of this paper is to persuade people to consider prior/posterior networks/EDL---see e.g. first two sentences in the abstract. While I think this is a worthy goal, the execution is underwhelming. E.g., there is no extensive comparison between EDL class of networks with standard UQ methods like deep ensemble and BNNs.

---

> ### Author Response · Authors · 2022-08-02
> **Response to Reviewer jdKi**
>
> We would like to thank reviewer jdKi for their helpful thoughts on our work, and much appreciate their suggestions on improvement.
> We integrated their minor improvements and addressed some of the more major concerns in the responses to the other reviewers in more detail.
> Overall, we welcome the idea to extend sections 3 and 4 to elaborate more on the different approaches, and we will dedicate a more detailed discussion on the nature epistemic
> uncertainty through EDL and BMA and to discuss the differences between EDL and other approaches such as ensembling.
>
> We also use P and p for discrete and continuous probability distributions, respectively, following suggested TMLR notational guidelines. We incorporated the other suggestions into the draft.

---

### Review · Reviewer_wKPZ · 2022-07-23

**Summary Of Contributions:**

This is a survey paper that summarizes works in the area called `evidential deep learning` (EDL) which is a sub-field of Bayesian deep learning. The main contribution of the survey paper including, a brief review the EDL approach to uncertainty estimates, collecting relevant papers, comparing different approaches. In additions, the authors also did a great job in displaying the difference/similarities of many recent works in several tables. As EDL is also new to me, I will use my best judgement and evaluate the survey mainly based on clarity and accessibility to readers who are unfamiliar with EDL.


**Requested Changes:**

In addition to the above points, I also have several questions and hope the authors can clarify them.

- From eq 5, the probability model (EDL), by default, assumes NO model uncertainty as the parameter $\hat \theta$ is deterministic. However, Sec 3.2 have discussions about how to obtain model (epistemic) uncertainty. This seems to contradict the assumptions of EDL. Are these estimates of heuristic nature, and are they not making sense from a Bayesian perspective?

- Can you formally define or describe what are evidential deep learning and Dirichlet networks? I couldn't find such definitions in the paper. The relation between Dirichlet networks and EDL is missing. Why are the Dirichlet networks so important here?

- I would suggest moving the Dirichlet distribution in Sec 2.1 to the beginning of Sec 3. In addition, the authors may consider spending some time detailing the Bayesian averaging, giving one or two concrete methods (e.g., dropout), and briefly mentioning the pros & cons. After that, define EDL formally and highlight the key ideas and benefits.

- As a newbie to EDL, I really want to see a concrete learning problem with concrete numbers/plots/explanations from several representative approaches to walk me through key innovations of EDL.

Other questions and comments.

- The right-hand sides of eq 7 & 9 have no $y$ while the left-hand sides contain the label $y$. It seems the entropy is computed for $p(y)$ rather than for $p(y|\mu)$. Are these quantities label-agnostic?

- Since you are using the `approximation` $P(y|x, D) \approx P(y|x, \hat \theta)$, isn't that all relevant equations should use $\approx$ instead of $=$?

- Eq 2 missing the requirement $\sum \mu_k =1 $? Does the measure $d \mathbf{ \mu}$ (in the appendix, e.g., eq 10) also have this constraint?




**Strengths And Weaknesses:**


## Strengths
- The overall structure of the paper seems reasonable, although I have a couple of concerns (see below.)
- The list of relevant work seems to be quite comprehensive.
- I like the tables in the paper, which are extremely helpful in visualizing the existing approaches and the deltas between

## Weakness
- The sub-structure of the paper does not seem very friendly to readers (like me) unfamiliar with EDL. I would appreciate it if the authors could provide formal descriptions/definitions of EDL, highlighting the key ideas and deltas (compared to model averaging.) Currently, I could not get the big idea of EDL.

- Structure in Sec 2 looks confusing. The authors introduce low-level technical details (Dirichlet distribution in Sec 2.1) before the high-level methodology, i.e., EDL, making the paper difficult to follow.

- The paper provides a high-level overview of existing works, lacking concrete examples. I would appreciate it if the authors could set up the stage for a concrete learning problem (e.g., supervised learning). This could be particularly useful for readers who are not native Bayesian. Otherwise, the paper is quite dry and difficult to follow.

---

> ### Author Response · Authors · 2022-08-02
> **Response to Reviewer wKPZ**
>
> We thank the anonymous reviewer for their constructive feedback and address some of the mentioned points:
>
> - EDL, with the exceptions mentioned in footnote 4, usually does not quantify epistemic uncertainty through
> the (trainable) network parameters and the bayesian model averaging as done by for instance ensembling.
> Instead, epistemic uncertainty is quantified through the uncertainty of the prior / posterior parameters (e.g., the concentration
> parameters of the prior or posterior Dirichlet). This could still be seen as a Bayesian approach to epistemic uncertainty, even if
> the uncertainty is quantified in a different way.
> - The term "Evidential Deep Learning" was originally taken from the Dempster-Shafer theory of evidence by Sensoy et al. (2018) (see section 3.3.2).
> In our work, we use the term broadly as a class of models that falls back onto a uniform prior when faced with a lack of evidence for a specific prediction.
> We will make this definition more clear and introduce it at an earlier point in the paper. Dirichlet networks are a natural choice within this
> framework for a classification problem, since the Dirichlet distribution is a conjugate prior for the categorical distribution, and thus a uniform Dirichlet
> is able to express a lack of evidence (assigning equal probability to all possible categorical distributions, which is different from a uniform categorical distribution assigning equal likelihood to all classes).
> - Based on the other reviewer's comment, we will expand the paper to discuss the difference with BMA in more detail and reposition the introduction of the Dirichlet in the paper.
>
> Responding to the other comments:
> - Eq. 7 & 9 might seem a bit confusing in the sense that y here does not refer to the concrete label in the dataset, but the random variable y. We would ask the reviewer to clarify this point in case we misunderstood their question.
> - We added the approximation sign addressing this note in eq. 5.
> - We added the missing constraint to the draft.

---

### Review · Reviewer_6GHq · 2022-07-25

**Summary Of Contributions:**

This survey paper looks at Evidential Deep Learning methods, mentioning and relating many different papers, as summarised in Figure 1 and Tables 1, 2 and 3. This includes methods for classification and regression, as well as prior-based and posterior-based methods.

**Broader Impact Concerns:**

No Broader Impact Statement provided, and I do not think one is required.

**Requested Changes:**

Many of my requested changes are detailed in the Weaknesses section above.

1. My biggest problem with the paper was my inability to follow the brief explanations of methods from Section 3.3 onwards. As a survey paper, I think the authors should have better writing so that I can follow each method and understand the big picture view of how they differ from each other. [See Weakness 1]

2. The authors claim they compare approaches to established methods, and also critically reflect on their advantages and shortcomings. Similar to point 1 above, I did not think this was done in a readable way or in sufficient detail. [Also see weakness 2]

3. [See Weakness 3] I did not follow what the most central theoretical results are. Could the authors make this more obvious and/or argue to me why this is a useful contribution. (Note that I have not read most of the surveyed papers, so am willing to be convinced that they are unaware of each other or use significantly different notation. But if this is the case, then perhaps the authors should make this clearer in the writing.)

4. See Weakness 4 and suggested change.

5. See Weakness 5 and suggested change.

6. I thought the Discussion brought up some valid problems with methods, but again, this was mentioned briefly and only at the end of the paper. Could some of these ideas be put in during the rest of the paper and in significantly more detail, so the methods are critically discussed throughout?

Minor typos: (a) 'i.e.' should be 'e.g.' bottom of page 2. (b) "probability *of* success" (bottom of page 2). (c) "a more complex teacher network" [no 's'] (top of page 7). (d) Greek letter 'nu' instead of 'v' in $\mathcal{N}(\gamma, \sigma^2 v^{-1})$ bottom of page 9? (e) "quanitified" -> "quantified" top of page 10.

**Strengths And Weaknesses:**

Strengths

1. The paper covers many different papers / methods.

2. Sections 1, 2, 3.1 and 3.2 are well-written, and easy to follow. I also thought the Discussion brought up some nice points.

Weaknesses

1. I found the explanations of the methods very brief and terse in Sections 3.3, 3.4 and 4. There was typically a couple of sentences assigned to each paper, and I was not able to follow how each method was different. This starts at the top of page 7. It seems like the authors assume that the reader is familiar with the papers, and this is just a brief summary of the methods like what would be in a space-pressured 'Related Works' section in a paper. As someone who does not know most of the recent works discussed in this survey paper, I was unable to follow most of the explanations.

2. The authors claim they compare Evidential Deep Learning approaches to established methods (as said at the top of page 2). Where was this done? Was this Section 5, and if so, I did not find this to be much of a discussion (it is just a few quick sentences on other approaches without much insight).

3. The authors say they provide the "most central theoretical results using a unified notation" (Abstract). This was not clear to me in the main paper at all. Most methods are mentioned but I am unsure what the most central results are and where in the paper they are. Most of the equations appear in the background section, and I believe most surveyed papers would already describe Dirichlet distributions.

4. The discussion on Model uncertainty in Section 3.2 was a bit weird to me. The authors previously argued that we lost epistemic uncertainty by using a single estimate of neural network weights, but now say that the uncertainty is still available. I think the authors should change some of the writing around Equation 4 so this does not seem so out-of-place / contradictory.

5. Figure 2 is not explained in much detail. I was often confused as to how each of the subplots in Figure 2 represents what the authors say it represents, and nowhere is this explained to the reader (instead, the authors just assume knowledge of eg how Fig 2d represents model uncertainty while 2c represents data uncertainty). Ideally, in my opinion, the authors should have a detailed caption (or detail in Section 2) explaining each of the subplots and comparing them to each other.

---

> ### Author Response · Authors · 2022-08-02
> **Reponse to Reviewer 6GHq**
>
> We would like to thank the anonymous reviewer for their tremendously useful feedback to our work.
> We corrected the caught typos and implemented a lot of the suggestions to further
> improve the paper. We address a few more of the raised points here:
>
> - Based on the feedback of the other reviewers, we welcome the idea
> of further discussing the showcased approaches in sections 3 and 4, which so far happened
> in an abridged version to stay within the 12 page limit.
> - We changed the wording of comparing EDL approaches to established methods in order not to imply
> new empirical results, which might have been misleading.
> The comparison indeed takes place within the discussion, and we will take this point along
> with the concerns of reviewer jdKi to extend the discussion of this point in the main text.
> - Addressing weakness 4 as well as a point raised by reviewer jdKi, we will add an additional section
> discussion the ways epistemic uncertainty are obtained within the EDL framework and how this compares to other
> UQ approaches such as ensembling.

---

### Author Response · Authors · 2022-08-19
**Updated submission available**

We just uploaded an updated submission to address the reviewers' concerns. Among smaller fixes and improvements, we also added the following sections in order to incorporate the feedback we received:

* Adding a definition of the term Evidential Deep Learning in section 2.3.
* Adding a note on epistemic uncertainty estimation in section 3.2.
* Dramatically extending the content in sections 3.3.1, 3.3.2, and 4.
* Extending the discussion in section 6.

---

### Decision · Action_Editors · 2022-08-31

**Recommendation:** Reject

**Comment:**

This paper presents a survey of evidential deep learning methods for uncertainty estimation in deep neural network classification.  This encompasses a collection of methods that develop the use of the Dirichlet distribution to model the outputs of the classifier.  The Dirichlet provides a natural formulation to balance new evidence with a prior over categorical outputs, and thus has advantages for uncertainty estimation and dealing with out-of-distribution data.

The reviewers found the paper interesting, of appropriate relevance to the community and timely, i.e. a well composed survey about evidential deep learning would be a useful contribution to the community.  They found that the survey seems to capture the breadth of the literature on this topic well.

There was substantial discussion relating to the paper.  In particular, the reviewers all found that the background and explanations of methods were too short and "terse" to serve as an adequate primer or introduction to the area.  The concern was essentially that in order to understand the survey, one had to already be intimately familiar with the methods, which of course is somewhat against the purpose of a survey paper.  Much of this issue seemed to stem from the authors' attempts to maintain a 12 page limit.  Reviewers asked authors to expand the explanations and welcomed going beyond 12 pages to do so.  At least one reviewer asked suggested adding open-source code would make the paper stronger and increase it's impact.

It seems that the authors agreed to expand on the paper and address the reviewers concerns.  They have indeed uploaded a new and significantly longer manuscript that has expanded explanations of the fundamental concepts and underlying methods and addressed other reviewer concerns.  However, this version was uploaded only shortly before reviewer decisions were due, and reviewers felt they did not have adequate time to re-review the new version of the paper.  As the reviewers felt uncomfortable accepting the paper without carefully reading the new version, the decision recommendation of the reviewers was reject, reject and leaning reject.

Therefore, I would recommend a reject.  However, I would strongly encourage that the authors resubmit, taking all the all the reviewer feedback into account, and ask for the same panel of action editor and reviewers (the reviewers have acknowledged that they would be willing to review the new manuscript in the context of these reviews).  The reviewers would like an opportunity to check the new version, but felt that it would have to reflect major revisions, and thus require more time than is allocated.